# Vocabulary In-Context Learning in Transformers: Benefits of Positional Encoding

**Qian Ma** [1]  **Ruoxiang Xu** [1]  **Yongqiang Cai** [1]*
[1]School of Mathematical Sciences, Beijing Normal University

## Abstract

Numerous studies have demonstrated that the Transformer architecture possesses the capability for in-context learning (ICL). In scenarios involving function approximation, context can serve as a control parameter for the model, endowing it with the universal approximation property (UAP). In practice, context is represented by tokens from a finite set, referred to as a vocabulary, which is the case considered in this paper, *i.e.*, vocabulary in-context learning (VICL). We demonstrate that VICL in single-layer Transformers, without positional encoding, does not possess the UAP; however, it is possible to achieve the UAP when positional encoding is included. Several sufficient conditions for the positional encoding are provided. Our findings reveal the benefits of positional encoding from an approximation theory perspective in the context of ICL.

## 1 Intruduction

Transformers have emerged as a dominant architecture in deep learning over the past few years. Thanks to their remarkable performance in language tasks, they have become the preferred framework in the natural language processing (NLP) field. A major trend in modern NLP is the development and integration of various black-box models, along with the construction of extensive text datasets. In addition, improving model performance in specific tasks through techniques such as in-context learning (ICL) [1, 2], chain of thought (CoT) [3, 4], and retrieval-augmented generation (RAG) [5] has become a significant research focus. While the practical success of these models and techniques is well-documented, the theoretical understanding of why they perform so well remains incomplete.

To explore the capabilities of Transformers in handling ICL tasks, it is essential to examine their approximation power. The universal approximation property (UAP) [6–9] has long been a key topic in the theoretical study of neural networks (NNs), with much of the focus historically on feed-forward neural networks (FNNs). Yun et al. [10] was the first to investigate the UAP of Transformers, demonstrating that any sequence-to-sequence function could be approximated by a Transformer network with fixed positional encoding. Luo et al. [11] highlighted that a Transformer with relative positional encoding does not possess the UAP. Meanwhile, Petrov et al. [12] explored the role of prompting in Transformers, proving that prompting a pre-trained Transformer can act as a universal functional approximator.

However, one limitation of these studies is that, in practical scenarios, the inputs to language models are derived from a finite set embedded in high-dimensional Euclidean space – commonly referred to as a vocabulary. Whether examining the work on prompts in [12] or the research on ICL in [13, 14], these studies assume inputs from the entire Euclidean space, which differs significantly from the discrete nature of vocabularies used in real-world applications.

---

*Email: caiyq.math@bnu.edu.cn. The first two authors made equal contributions to the paper.

39th Conference on Neural Information Processing Systems (NeurIPS 2025).

## 1.1 Contributions

Starting with the connection between FNNs and Transformers, we turn to the finite restriction of vocabularies and study the benefits of positional encoding. Leveraging the UAP of FNNs, we explore the UAP of Transformers for ICL tasks in two scenarios: one where positional encoding is used and one where it is not. In both cases, the inputs are from a finite vocabulary. More specifically:

1. We first establish a connection between FNNs and Transformers in processing ICL tasks (Lemma 3). Using this lemma, we show that Transformers can function as universal approximators (Lemma 4), where the context serves as control parameters, while the weights and biases of the Transformer remain fixed.

2. When the vocabulary is finite and positional encoding is not used, we prove that single-layer Transformers cannot achieve the UAP for ICL tasks (Theorem 6).

3. However, when positional encoding is used, it becomes possible for single-layer Transformers to achieve the UAP (Theorem 7). In particular, for Transformers with ReLU or softmax activation functions, the conditions on the positional encoding are relaxed (Theorem 8).

## 1.2 Related Works

**Universal approximation property.** NNs, through multi-layer nonlinear transformations and feature extraction, are capable of learning deep feature representations from raw data. As neural networks gain prominence, theoretical investigations—especially into their UAP – have intensified. Related studies typically fall into two categories: those allowing arbitrary width with fixed depth [6–9], and those allowing arbitrary depth with bounded width [15–19]. Since our study builds on existing results regarding the approximation capabilities of FNNs, we focus on investigating the approximation abilities of single-layer Transformers in modulating context for ICL tasks. Consequently, our work relies more on the findings from the first category of research. The realization of the UAP depends on the architecture of the network itself, providing constructive insights for exploring the connection between FNNs and Transformers. Recently, Petrov et al. [12] also explored UAP in the context of ICL, but without considering vocabulary constraints or positional encodings.

**Transformers.** The Transformer is a widely used neural network architecture for modeling sequences [20–25]. This non-recurrent architecture relies entirely on the attention mechanism to capture global dependencies between inputs and outputs [20]. The neural sequence transduction model is typically structured using an encoder-decoder framework [26, 27].

Without positional encoding, the Transformer can be viewed as a stack of $N$ blocks, each consisting of a self-attention layer followed by a feed-forward layer with skip connections. In this paper, we focus on the case of a single-layer self-attention sequence encoder.

**In-context learning.** The Transformer has demonstrated remarkable performance in the field of NLP, and large language models (LLMs) are gaining increasing popularity. ICL has emerged as a new paradigm in NLP, enabling LLMs to make better predictions through prompts provided within the context [2, 28–31]. ICL delivers high performance with high-quality data at a lower cost [32–34]. It enhances retrieval-augmented methods by prepending grounding documents to the input [35] and can effectively update or refine the model's knowledge base through well-designed prompts [36].

**Positional Encoding.** The following explanation clarifies the significance of incorporating positional encoding into the Transformer architecture. RNNs capture sequential order by encoding the changes in hidden states over time. In contrast, for Transformers, the self-attention mechanism is permutation equivariant, meaning that for any model $f$, any permutation matrix $\pi$, and any input $x$, the following holds: $f(\pi(x)) = \pi(f(x))$.

We aim to explore the impact of positional encoding on the performance of a single-layer Transformer when performing ICL tasks with a finite vocabulary. Therefore, we focus on analyzing existing positional encoding methods. There are fundamental methods for encoding positional information in a sequence within the Transformer: absolute positional encodings (APEs), *e.g.* [37, 25, 38, 39], relative positional encodings (RPEs), *e.g.* [40, 41, 39] and rotary positional embedding (RoPE) [42]. The commonly used APE is implemented by directly adding the positional encodings to the word embeddings, and we follow this implementation.

**UAP of ICL.** To understand the mechanism of ICL, various explanations have been proposed, including those based on Bayesian theory [43, 44] and gradient descent theory [45]. Fine-tuning the Transformer through ICL alters the presentation of the input rather than the model parameters, which is driven by successful few-shot and zero-shot learning [46, 47]. This success raises the question of whether we can achieve the UAP through context adjustment.

Yun et al. [10] demonstrated that Transformers can serve as universal sequence-to-sequence approximators, while Alberti et al. [48] extended the UAP to architectures with non-standard attention mechanisms. However, their implementations allow the internal parameters of the Transformers to vary, which does not fully capture the nature of ICL. In contrast, Likhosherstov et al. [49] showed that while the parameters of self-attention remain fixed, various sparse matrices can be approximated by altering the inputs. Fixing self-attention parameters aligns more closely with practical scenarios and provides valuable insights for our work. However, this approach has the limitation of excluding the full Transformer architecture. Furthermore, Deora et al. [50] illustrated the convergence and generalization of single-layer multi-head self-attention models trained using gradient descent, supporting the feasibility of our research by emphasizing the robust generalization of Transformers. Nevertheless, Petrov et al. [51] indicated that the presence of a prefix does not alter the attention focus within the context, prompting us to explore variations in input context and introduce flexibility in positional encoding.

## 1.3 Outline

We will introduce the notations and background results in Section 2. Section 3 addresses the case where the vocabulary is finite and positional encoding is not used. Section 4 discusses the benefits of using positional encoding. A summary is provided in Section 5. All proofs of lemmas and theorems are provided in the Appendix.

## 2 Background Materials

We consider the approximation problem as follows. Given a fixed Transformer network, for any target continuous function $f : \mathcal{K} \to \mathbb{R}^{d_y}$ with a compact domain $\mathcal{K} \subset \mathbb{R}^{d_x}$, we aim to adjust the content of the context so that the output of the Transformer network can approximate $f$. First, we present the concrete forms and notations for the inputs of ICL, FNNs, and Transformers.

### 2.1 Notations

**Input of in-context learning.** In the ICL task, the given $n$ demonstrations are denoted as $z^{(i)} = (x^{(i)}, y^{(i)})$ for $i = 1, 2, ..., n$, where $x^{(i)} \in \mathbb{R}^{d_x}$ and $y^{(i)} \in \mathbb{R}^{d_y}$. Unlike the setting in [13, 14] where $y^{(i)}$ was related to $x^{(i)}$ (for example $y^{(i)} = \phi(x^{(i)})$ for some function $\phi$), we do not assume any correspondence between $x^{(i)}$ and $y^{(i)}$, *i.e.*, $x^{(i)}$ and $y^{(i)}$ are chosen freely. To predict the target at a query vector $x \in \mathbb{R}^{d_x}$ or $z = (x, 0) \in \mathbb{R}^{d_x+d_y}$, we define the input matrix $Z$ as follows:

$$Z = \begin{bmatrix} z^{(1)} & z^{(2)} & \cdots & z^{(n)} & z \end{bmatrix} := \begin{bmatrix} x^{(1)} & x^{(2)} & \cdots & x^{(n)} & x \\ y^{(1)} & y^{(2)} & \cdots & y^{(n)} & 0 \end{bmatrix} \in \mathbb{R}^{(d_x+d_y) \times (n+1)}. \quad (1)$$

Furthermore, let $\mathcal{P} : \mathbb{N}^+ \to \mathbb{R}^{d_x+d_y}$ represent a positional encoding function, and define $\mathcal{P}^{(i)} := \mathcal{P}(i)$. Denote the demonstrations with positional encoding as $z_{\mathcal{P}}^{(i)} := z^{(i)} + \mathcal{P}^{(i)}$ and $z_{\mathcal{P}} := z + \mathcal{P}^{(n+1)}$. The context with positional encoding can then be represented as:

$$Z_{\mathcal{P}} = \begin{bmatrix} z_{\mathcal{P}}^{(1)} & z_{\mathcal{P}}^{(2)} & \cdots & z_{\mathcal{P}}^{(n)} & z_{\mathcal{P}} \end{bmatrix} := \begin{bmatrix} x_{\mathcal{P}}^{(1)} & x_{\mathcal{P}}^{(2)} & \cdots & x_{\mathcal{P}}^{(n)} & x_{\mathcal{P}} \\ y_{\mathcal{P}}^{(1)} & y_{\mathcal{P}}^{(2)} & \cdots & y_{\mathcal{P}}^{(n)} & y_{\mathcal{P}} \end{bmatrix} \in \mathbb{R}^{(d_x+d_y) \times (n+1)}. \quad (2)$$

Additionally, we denote:

$$X = \begin{bmatrix} x^{(1)} & x^{(2)} & \cdots & x^{(n)} \end{bmatrix} \in \mathbb{R}^{d_x \times n}, \quad X_{\mathcal{P}} = \begin{bmatrix} x_{\mathcal{P}}^{(1)} & x_{\mathcal{P}}^{(2)} & \cdots & x_{\mathcal{P}}^{(n)} \end{bmatrix} \in \mathbb{R}^{d_x \times n}, \quad (3)$$

$$Y = \begin{bmatrix} y^{(1)} & y^{(2)} & \cdots & y^{(n)} \end{bmatrix} \in \mathbb{R}^{d_y \times n}, \quad Y_{\mathcal{P}} = \begin{bmatrix} y_{\mathcal{P}}^{(1)} & y_{\mathcal{P}}^{(2)} & \cdots & y_{\mathcal{P}}^{(n)} \end{bmatrix} \in \mathbb{R}^{d_y \times n}. \quad (4)$$

**Feed-forward neural networks.**    One-hidden-layer FNNs have sufficient capacity to approximate continuous functions on any compact domain. In this article, all the FNNs we refer to and use are one-hidden-layer networks. We denote a one-hidden-layer FNN with activation function $\sigma$ as $\mathrm{N}^\sigma$, and the set of all such networks is denoted as $\mathcal{N}^\sigma$, *i.e.*,

$$\mathcal{N}^\sigma = \left\{ \mathrm{N}^\sigma := A\,\sigma(Wx+b) \,\middle|\, A \in \mathbb{R}^{d_y \times k},\ W \in \mathbb{R}^{k \times d_x},\ b \in \mathbb{R}^k,\ k \in \mathbb{N}^+ \right\}$$

$$= \left\{ \mathrm{N}^\sigma := \sum_{i=1}^k a_i \sigma(w_i \cdot x + b_i) \,\middle|\, (a_i, w_i, b_i) \in \mathbb{R}^{d_y} \times \mathbb{R}^{d_x} \times \mathbb{R},\ k \in \mathbb{N}^+ \right\}. \tag{5}$$

For element-wise activations, such as ReLU, the above notation is well-defined. However, for non-element-wise activation functions, which are not widely used but considered in this article, especially the softmax activation, we need to give more details for the notation:

$$\mathcal{N}^{\text{softmax}} = \left\{ \mathrm{N}^{\text{softmax}} = \frac{\sum_{i=1}^k a_i \mathrm{e}^{w_i \cdot x + b_i}}{\sum_{i=1}^k \mathrm{e}^{w_i \cdot x + b_i}} \,\middle|\, (a_i, w_i, b_i) \in \mathbb{R}^{d_y} \times \mathbb{R}^{d_x} \times \mathbb{R},\ k \in \mathbb{N}^+ \right\}. \tag{6}$$

**Transformers.**    We define the general attention mechanism following [13, 14] as:

$$\mathrm{Attn}^\sigma_{Q,K,V}(Z) := VZM\sigma\big((QZ)^\top KZ\big), \tag{7}$$

where $V$, $Q$, $K$ are the value, query, and key matrices in $\mathbb{R}^{(d_x+d_y) \times (d_x+d_y)}$, respectively. $M = \mathrm{diag}(I_n, 0)$ is the mask matrix in $\mathbb{R}^{(n+1) \times (n+1)}$, and $\sigma$ is the activation function. Note that the context vectors $z^{(i)}$ are asymmetric and do not include the query vector $z$ itself; therefore, we introduce a mask matrix $M$ following the design of [13, 14]. Here the softmax activation of a matrix $G \in \mathbb{R}^{m \times n}$ is defined as:

$$\big(\mathrm{softmax}(G)\big)_{i,j} := \frac{\exp\big(G_{i,j}\big)}{\sum_{l=1}^m \exp\big(G_{l,j}\big)}. \tag{8}$$

With this formulation of the general attention mechanism, we can define a single-layer Transformer without positional encoding as:

$$\mathrm{T}^\sigma(x; X, Y) := (Z + VZM\sigma((QZ)^\top KZ))_{d_x+1:d_x+d_y,\,n+1}, \tag{9}$$

where $[a:b, c:d]$ denotes the submatrix from the $a$-th row to the $b$-th row and from the $c$-th column to the $d$-th column. If $a = b$ (or $c = d$), the row (or column) index is reduced to a single number. Similarly to the notation for FNNs, $\mathcal{T}^\sigma$ denotes the set of all $\mathrm{T}^\sigma$ with different parameters.

**Vocabulary.**    In the above notations, the tokens in context of ICL are general and unrestricted. When we refer to a "vocabulary", we mean that the tokens are drawn from a finite set. More specifically, we refer to it as VICL if all input vectors $z^{(i)}$ come from a finite vocabulary $\mathcal{V} = \mathcal{V}_x \times \mathcal{V}_y \subset \mathbb{R}^{d_x} \times \mathbb{R}^{d_y}$. In this case, we use subscript $*$, *i.e.*, $\mathrm{T}^\sigma_*(x; X, Y)$, to represent the Transformer $\mathrm{T}^\sigma(x; X, Y)$ defined in equation (9), and denote the set of such Transformers as $\mathcal{T}^\sigma_*$:

$$\mathcal{T}^\sigma_* = \left\{ \mathrm{T}^\sigma_*(x; X, Y) := \mathrm{T}^\sigma(x; X, Y) \,\middle|\, z^{(i)} \in \mathcal{V},\ i \in \{1, 2, \cdots, n\},\ n \in \mathbb{N}^+ \right\}. \tag{10}$$

To facilitate the simplification of VICL analysis, we denote a FNN with a finite set of weights as $\mathrm{N}^\sigma_*$, and the corresponding set of such networks as $\mathcal{N}^\sigma_*$. More specifically, when the activation function is an elementwise activation, we denote:

$$\mathcal{N}^\sigma_* = \left\{ \mathrm{N}^\sigma_* := \sum_{i=1}^k a_i \sigma(w_i \cdot x + b_i) \,\middle|\, (a_i, w_i, b_i) \in \mathcal{A} \times \mathcal{W} \times \mathcal{B},\ k \in \mathbb{N}^+ \right\}, \tag{11}$$

where $\mathcal{A} \subset \mathbb{R}^{d_y}$, $\mathcal{W} \subset \mathbb{R}^{d_x}$, and $\mathcal{B} \subset \mathbb{R}$ are finite sets. When the activation function is softmax, we denote:

$$\mathcal{N}^{\text{softmax}}_* = \left\{ \mathrm{N}^{\text{softmax}} = \frac{\sum_{i=1}^k a_i e^{w_i \cdot x + b_i}}{\sum_{i=1}^k e^{w_i \cdot x + b_i}} \,\middle|\, (a_i, w_i, b_i) \in \mathcal{A} \times \mathcal{W} \times \mathcal{B},\ k \in \mathbb{N}^+ \right\}, \tag{12}$$

where $\mathcal{A}, \mathcal{W}$ and $\mathcal{B}$ are defined as in the previous context.

**Positional encoding.** When positional encoding $\mathcal{P}$ is involved, we add the subscript $\mathcal{P}$, *i.e.*,

$$\mathcal{T}_{*,\mathcal{P}}^{\sigma} = \left\{ \mathrm{T}_{*,\mathcal{P}}^{\sigma}(x; X, Y) := \mathrm{T}^{\sigma}(x_{\mathcal{P}}; X_{\mathcal{P}}, Y_{\mathcal{P}}) \mid z^{(i)} \in \mathcal{V}, i \in \{1, 2, ..., n\}, n \in \mathbb{N}^{+} \right\}. \tag{13}$$

Note that the context length $n$ in $\mathrm{T}^{\sigma}$, $\mathrm{T}_{*}^{\sigma}$ and $\mathrm{T}_{*,\mathcal{P}}^{\sigma}$ are unbounded.

We present all our notations in Table 1 in Appendix A for easy reference.

## 2.2 Universal Approximation Property

The vanilla form of the UAP for FFNs plays a crucial role in our study. Before we state this property as a formal lemma, we put forward the following assumption first, which is similar to the one in [14] and is used to simplify the analysis of Transformers.

**Assumption 1.** *The matrices $Q$, $K$, $V \in \mathbb{R}^{(d_x+d_y) \times (d_x+d_y)}$ have the following sparse partition:*

$$Q = \begin{bmatrix} B & 0 \\ 0 & 0 \end{bmatrix}, \quad K = \begin{bmatrix} C & 0 \\ 0 & 0 \end{bmatrix}, \quad V = \begin{bmatrix} D & E \\ F & U \end{bmatrix}, \tag{14}$$

*where $B$, $C$, $D \in \mathbb{R}^{d_x \times d_x}$, $E \in \mathbb{R}^{d_x \times d_y}$, $F \in \mathbb{R}^{d_y \times d_x}$ and $U \in \mathbb{R}^{d_y \times d_y}$. Furthermore, the matrices $B$, $C$ and $U$ are non-singular, and the matrix $F = 0$.*

In addition, we assume the element-wise activation $\sigma$ is non-polynomial, locally bounded, and continuous. It is worth noting that a randomly initialized $n \times n$ matrix is non-singular with probability one, so it is acceptable to assume that the matrices $B, C$ and $U$ are non-singular. Moreover, the assumption $F = 0$ can be relaxed, which will be discussed in Appendix F. Here, we have slightly strengthened it for the sake of computational convenience.

**Lemma 2** (UAP of FNNs [9])**.** *Let $\sigma : \mathbb{R} \to \mathbb{R}$ be a non-polynomial, locally bounded, piecewise continuous activation function. For any continuous function $f : \mathbb{R}^{d_x} \to \mathbb{R}^{d_y}$ defined on a compact domain $\mathcal{K}$, and for any $\varepsilon > 0$, there exist $k \in \mathbb{N}^{+}$, $A \in \mathbb{R}^{d_y \times k}$, $b \in \mathbb{R}^{k}$, and $W \in \mathbb{R}^{k \times d_x}$ such that*

$$\|A\sigma(Wx + b) - f(x)\| < \varepsilon, \quad \forall x \in \mathcal{K}. \tag{15}$$

The theorem presented above is well-known and primarily applies to activation functions operating element-wise. However, it can be readily extended to the case of the softmax activation function. In fact, this can be achieved using NNs with exponential activation functions. The specific approach for this generalization is detailed in Appendix B.

## 2.3 Feed-forward neural networks and Transformers

It is important to emphasize the connection between FNNs and Transformers, which will be represented in the following lemmas and are crucial for establishing our main theory.

**Lemma 3.** *Let $\sigma : \mathbb{R} \to \mathbb{R}$ be a non-polynomial, locally bounded, piecewise continuous activation function, and $\mathrm{T}^{\sigma}$ be a single-layer Transformer satisfying Assumption 1. For any one-hidden-layer network $\mathrm{N}^{\sigma} : \mathbb{R}^{d_x-1} \to \mathbb{R}^{d_y} \in \mathcal{N}^{\sigma}$ with $n$ hidden neurons, there exist matrices $X \in \mathbb{R}^{d_x \times n}$ and $Y \in \mathbb{R}^{d_y \times n}$ such that*

$$\mathrm{T}^{\sigma}(\tilde{x}; X, Y) = \mathrm{N}^{\sigma}(x), \quad \forall x \in \mathbb{R}^{d_x-1}. \tag{16}$$

There is a difference in the input dimensions of $\mathrm{T}^{\sigma}$ and $\mathrm{N}^{\sigma}$, as the latter includes a bias dimension absent in the former. To connect the two inputs, $\tilde{x}$ and $x$, we use a tilde, where $\tilde{x}$ is formed by augmenting $x$ with an additional one appended to the end.

By employing the structure of $K$, $Q$ and $V$ in equation (14), the output forms of the Transformer $\mathrm{T}^{\sigma}(\tilde{x}; X, Y)$ can be simplified as follows:

$$\mathrm{T}^{\sigma}(\tilde{x}; X, Y) = \left( \begin{bmatrix} X & \tilde{x} \\ Y & 0 \end{bmatrix} + \begin{bmatrix} DX + EY & 0 \\ FX + UY & 0 \end{bmatrix} \sigma \left( \begin{bmatrix} X^{\top}B^{\top}CX & X^{\top}B^{\top}C\tilde{x} \\ \tilde{x}^{\top}B^{\top}CX & \tilde{x}^{\top}B^{\top}C\tilde{x} \end{bmatrix} \right) \right)_{d_x+1:d_x+d_y, n+1}$$

$$= (FX + UY)\sigma(X^{\top}B^{\top}C\tilde{x}) = UY\sigma(X^{\top}B^{\top}C\tilde{x}). \tag{17}$$

Comparing this with the output form of FNNs, *i.e.*, $\mathrm{N}^{\sigma}(x) = A\sigma(Wx + b)$, it becomes evident that setting $X = (C^{\top}B)^{-1}[W \quad b]^{\top}$ and $Y = U^{-1}A$ is sufficient to finish the proof.

It can be observed that the form in equation (17) exhibits the structure of an FNN. Consequently, Lemma 3 implies that single-layer Transformers $T^\sigma$ in the context of ICL and FNNs $N^\sigma$ are equivalent. However, this equivalence does not hold for the case of softmax activation due to differences in the normalization operations between FNNs and Transformers. Therefore, in the subsequent sections of this article, we employ different analytical methods to address the two types of activation functions.

Moreover, the equivalence in equation (16) suggests that the context in Transformers can act as a control parameter for the model, thereby endowing it with the UAP.

### 2.4 Universal Approximation Property of In-context Learning

We now present the UAP of Transformers in the context of ICL.

**Lemma 4.** *Let $\sigma : \mathbb{R} \to \mathbb{R}$ be a non-polynomial, locally bounded, piecewise continuous activation function or softmax function, and $T^\sigma$ be a single-layer Transformer satisfying Assumption 1, and $\mathcal{K}$ be a compact domain in $\mathbb{R}^{d_x-1}$. Then for any continuous function $f : \mathcal{K} \to \mathbb{R}^{d_y}$ and any $\varepsilon > 0$, there exist matrices $X \in \mathbb{R}^{d_x \times n}$ and $Y \in \mathbb{R}^{d_y \times n}$ such that*

$$\left\| T^\sigma(\tilde{x}; X, Y) - f(x) \right\| < \varepsilon, \quad \forall x \in \mathcal{K}. \tag{18}$$

For the case of element-wise activation, the result follows directly by combining Lemma 2 and Lemma 3. However, for the softmax activation, the normalization operation requires an additional technique in the proof. The core idea is to construct an FNN with exponential activation functions, incorporating an additional neuron to handle the normalization effect. Detailed proofs are provided in Appendix B. Similar results have been obtained in recent work [12], though via different methodologies.

## 3 The Non-Universal Approximation Property of $\mathcal{N}_*^\sigma$ and $\mathcal{T}_*^\sigma$

One key aspect of ICL is that the context can act as a control parameter for the model. We now consider the case where the tokens in context are restricted to a finite vocabulary. A natural question arises: can single-layer Transformers with a finite vocabulary, *i.e.*, $\mathcal{T}_*^\sigma$, still achieve the UAP via ICL? We first analyze $\mathcal{N}_*^\sigma$ for simplicity, and then use the established connection between FNNs and Transformers to extend the result to $\mathcal{T}_*^\sigma$. The answer is that $\mathcal{N}_*^\sigma$ cannot achieve the UAP because of the restriction of finite parameters.

For element-wise activations, the span of $\mathcal{N}_*^\sigma$, span$\{\mathcal{N}_*^\sigma\}$, forms a finite-dimensional function space. According to results from functional analysis, span$\{\mathcal{N}_*^\sigma\}$ is closed under the function norm (see e.g. Theorem 1.21 of [52] or Corollary C.4 of [53]). This implies that the set of functions that can be approximated by span$\{\mathcal{N}_*^\sigma\}$ is precisely the set of functions within span$\{\mathcal{N}_*^\sigma\}$. Consequently, any function not in span$\{\mathcal{N}_*^\sigma\}$ cannot be arbitrarily approximated, meaning that the UAP cannot be achieved.

For softmax networks, the normalization operation introduces further limitations. Even though $N_*^{\text{softmax}}$ consists of weighted units drawn from a fixed finite collection of basic units, normalization prevents these networks from being simple linear combinations of one another. While the span of $\mathcal{N}_*^{\text{softmax}}$ might theoretically have infinite dimensionality, its expressive power remains constrained.

To better understand the functional behavior of $\mathcal{N}_*^{\text{softmax}}$, we introduce the structural Proposition 12 whose details are provided in Appendix C. The proposition characterizes the maximum number of zero points that functions in this class can exhibit, and the result can be established via mathematical induction. This observation motivates the following lemma, which formally states the non-universal approximation property of $\mathcal{N}_*^\sigma$.

**Lemma 5.** *The function class $\mathcal{N}_*^\sigma$, with a non-polynomial, locally bounded, piecewise continuous element-wise activation function or softmax activation function $\sigma$, cannot achieve the UAP. Specifically, there exist a compact domain $\mathcal{K} \subset \mathbb{R}^{d_x}$, a continuous function $f : \mathcal{K} \to \mathbb{R}^{d_y}$, and $\varepsilon_0 > 0$ such that*

$$\max_{x \in \mathcal{K}} \left\| f(x) - N_*^\sigma(\tilde{x}) \right\| \geq \varepsilon_0, \quad \forall N_*^\sigma \in \mathcal{N}_*^\sigma. \tag{19}$$

In the proof of Lemma 5, we demonstrated through Proposition 12 that the number of zeros of $N_*^{\text{softmax}}$ depends solely on a finite set of parameters and constitutes a bounded quantity. Functions

can be explicitly constructed whose number of zeros exceeds this bound, thereby preventing their approximation within $\mathcal{N}_*^{\mathrm{softmax}}$.

By leveraging the connection between FNNs and Transformers, we establish Theorem 6 to demonstrate that $\mathcal{T}_*^{\sigma}$ cannot achieve the UAP.

**Theorem 6.** *The function class $\mathcal{T}_*^{\sigma}$, with a non-polynomial, locally bounded, piecewise continuous element-wise activation function or softmax activation function $\sigma$ and every $\mathrm{T}^{\sigma} \in \mathcal{T}_*^{\sigma}$ satisfies Assumption 1, cannot achieve the UAP. Specifically, there exist a compact domain $\mathcal{K} \subset \mathbb{R}^{d_x}$, a continuous function $f : \mathcal{K} \to \mathbb{R}^{d_y}$, and $\varepsilon_0 > 0$ such that*

$$\max_{x \in \mathcal{K}} \big\| f(x) - \mathrm{T}_*^{\sigma}(\tilde{x}) \big\| \geq \varepsilon_0, \quad \forall\, \mathrm{T}_*^{\sigma} \in \mathcal{T}_*^{\sigma}. \tag{20}$$

The result for element-wise activations follows directly from the application of Lemma 3 and Lemma 5. However, the case of the softmax activation requires additional techniques to account for the normalization effect. The proof, which utilizes Proposition 12 once again, is presented in the Appendix C. It is worth noting that Theorem 6 holds even without imposing any constraints on the $V$, $Q$ and $K$ (e.g., the sparse partition described in equation (14)). Further details can be found in Appendix F.

# 4   The Universal Approximation Property of $\mathcal{T}_{*,\mathcal{P}}^{\sigma}$

After establishing that neither $\mathcal{N}_*^{\sigma}$ nor $\mathcal{T}_*^{\sigma}$ can achieve the UAP, we aim to leverage a key feature of Transformers: their ability to incorporate APEs during token input. This motivates us to investigate whether $\mathcal{T}_{*,\mathcal{P}}^{\sigma}$ can realize the UAP.

The answer is affirmative. To support our constructive proof, we invoke the Kronecker Approximation Theorem as a key auxiliary tool, which will be stated as Lemma 13. This result ensures the density of certain structured sets in $\mathbb{R}^n$ under mild arithmetic conditions. The formal statement and discussion of this theorem are provided in Appendix D.

**Theorem 7.** *Let $\mathcal{T}_{*,\mathcal{P}}^{\sigma}$ be the class of functions $\mathrm{T}_{*,\mathcal{P}}^{\sigma}$ satisfying Assumption 1, with a non-polynomial, locally bounded, piecewise continuous element-wise activation function $\sigma$, the subscript refers to the finite vocabulary $\mathcal{V} = \mathcal{V}_x \times \mathcal{V}_y$, $\mathcal{P} = \mathcal{P}_x \times \mathcal{P}_y$ represents the positional encoding map, and denote a set $S$ as:*

$$S := \mathcal{V}_x + \mathcal{P}_x = \left\{ x_i + \mathcal{P}_x^{(j)} \ \middle|\ x_i \in \mathcal{V}_x,\ i,\ j \in \mathbb{N}^+ \right\}. \tag{21}$$

*If $S$ is dense in $\mathbb{R}^{d_x}$, $\{1,\ -1,\ \sqrt{2},\ 0\}^{d_y} \subset \mathcal{V}_y$ and $\mathcal{P}_y = 0$, then $\mathcal{T}_{*,\mathcal{P}}^{\sigma}$ can achieve the UAP. More specifically, given a network $\mathrm{T}_{*,\mathcal{P}}^{\sigma}$, then for any continuous function $f : \mathbb{R}^{d_x-1} \to \mathbb{R}^{d_y}$ defined on a compact domain $\mathcal{K}$ and $\varepsilon > 0$, there always exist $X \in \mathbb{R}^{d_x \times n}$ and $Y \in \mathbb{R}^{d_y \times n}$ from the vocabulary $\mathcal{V}$, i.e., $x^{(i)} \in \mathcal{V}_x, y^{(i)} \in \mathcal{V}_y$, with some length $n \in \mathbb{N}^+$ such that*

$$\big\| \mathrm{T}_{*,\mathcal{P}}^{\sigma}\left(\tilde{x}; X, Y\right) - f(x) \big\| < \varepsilon, \quad \forall x \in \mathcal{K}. \tag{22}$$

We provide a constructive proof in Appendix C, and here we only demonstrate the proof idea by considering the specific case of $d_y = 1$ and assuming the matrice $U$ in the Transformer $\mathrm{T}_{*,\mathcal{P}}^{\sigma}$ is an identity matrice. In this case, the Transformer can be simplified to an FNN $\mathrm{N}_*^{\sigma}$, that is

$$\mathrm{T}_{*,\mathcal{P}}^{\sigma}(x; X, Y) = Y_{\mathcal{P}} \sigma\big(X_{\mathcal{P}}^{\top} B^{\top} C \tilde{x}\big) = \sum_{j=1}^{n} y^{(j)} \sigma\bigg( \big(x^{(j)} + \mathcal{P}_x^{(j)}\big) B^{\top} C \cdot \tilde{x} \bigg), \tag{23}$$

which is similar to the calculation in equation (17). The UAP of FNNs shown in Lemma 2 implies that the target function $f$ can be approximated by an FNN with $k$ hidden neurons, that is

$$\mathrm{N}^{\sigma}(x) = A\sigma(W\tilde{x} + b) = \sum_{i=1}^{k} a_i \sigma(w_i \cdot x + b_i) = \sum_{i=1}^{k} a_i \sigma(\tilde{w}_i \cdot \tilde{x}). \tag{24}$$

Since we are considering a continuous activation function $\sigma$, we can conclude that slightly perturbing the parameters $A$ and $W$ will lead to new FNN that can still approximate $f$. This motivates us to construct a proof using the property that each $\tilde{w}_i \in \mathbb{R}^{d_x}$ can be approximated by vectors

$x_{\mathcal{P}} B^\top C, x_{\mathcal{P}} \in S = \mathcal{V}_x + \mathcal{P}_x$, and each $a_i \in \mathbb{R}$ can be approximated by $q_i \sqrt{2} \pm l_i$, with positive integers $q_i$ and $l_i$.

For ease of exposition, we will first show how to construct $X$ and $Y$, so as to approximate the first term in the summation in equation (24), namely $a_1 \sigma(\tilde{w}_1 \cdot \tilde{x})$. By Lemma 13, we may choose positive integers $q$ and $l$ such that $q\sqrt{2} \pm l$ is sufficiently close to $a_1$. Consider the first token in the context, since the positional encoding is fixed, *i.e.*, $\mathcal{V}_x + \mathcal{P}^{(1)}$ is a finite set, then one of two cases must occur:

1. if there exists a token $x^{(1)} \in \mathcal{V}_x$ for which $x^{(1)} + \mathcal{P}^{(1)}$ is sufficiently close to $\tilde{w}_1$, then we declare this position "valid";

2. otherwise, we declare the position "invalid", and choose any $x^{(1)} \in \mathcal{V}_x$, and set $y^{(1)} = 0$ so as to nullify its contribution in the sum.

We then proceed inductively: having handled the first token, we construct the second token in exactly the same manner, then the third, and so on, until we have identified $q + l$ valid positions. Because $S$ is dense in $\mathbb{R}^{d_x}$ and $q, l$ are finite, this selection process necessarily terminates after finitely many steps. Finally, we assign $y^{(i)} = \sqrt{2}$ for $q$ of the valid positions and $y^{(i)} = \pm 1$ for other $l$ valid positions. Up to now, we have built a partial context that enables the output of $\mathrm{T}^\sigma_{*,\mathcal{P}}$ to approximate $a_1 \sigma(\tilde{w}_1 \cdot \tilde{x})$ with arbitrarily small error. Once we have approximated $a_1 \sigma(\tilde{w}_1 \cdot \tilde{x})$, we can then approximate $a_2 \sigma(\tilde{w}_2 \cdot \tilde{x}), \cdots, a_k \sigma(\tilde{w}_k \cdot \tilde{x})$ in finitely many steps, thereby completing the construction of the full context $X$ and $Y$. In the proof idea above, we take the density of the set $S$ in $\mathbb{R}^{d_x}$ as a fundamental assumption. $\mathcal{V}_x$ contains only finitely many elements, rendering it bounded. For $S$ to be dense in the entire space, $\mathcal{P}_x$ must be unbounded. We further extend the above approach to discuss whether other forms of positional encoding can also achieve the UAP in Appendix E.4.

Next, we relax this requirement, eliminating the need for $\mathcal{P}_x$ to be unbounded, making the conditions more aligned with practical scenarios. Particularly, we consider the specific activation function in the following theorem, where the notations not explicitly mentioned remain consistent with those in Theorem 7. We present an informal version, and the formal version is provided in Appendix E.

**Theorem 8** (Informal Version). *If the set $S$ is dense in $[-1, 1]^{d_x}$, then $\mathcal{T}^{\mathrm{ReLU}}_{*,\mathcal{P}}$ is capable of achieving the UAP. Additionally, if $S$ is only dense in a neighborhood $B(w^*, \delta)$ of a point $w^* \in \mathbb{R}^{d_x}$ with radius $\delta > 0$, then the class of transformers with exponential activation, i.e., $\mathcal{T}^{\exp}_{*,\mathcal{P}}$, is capable of achieving the UAP.*

The density condition on $S$ is significantly refined here, which we will discuss in the later remark. This improvement is possible because the proof of Theorem 7 relies directly on the UAP of FNNs, where the weights take values from the entire parameter space. However, for FNNs with specific activations, we can restrict the weights to a small set without losing the UAP.

For ReLU networks, we can use the positive homogeneity property, *i.e.*, $A\mathrm{ReLU}(W\tilde{x}) = \lambda^{-1} A\mathrm{ReLU}(\lambda W\tilde{x})$ for any $\lambda > 0$, to restrict the weight matrix $W$. In fact, the restriction that all elements of $W$ take values in the interval $[-1, 1]$ does not affect the UAP of ReLU FNNs because the scale of $W$ can be recovered by adjusting the scale of $A$ via choosing a proper $\lambda$.

For exponential networks, the condition on $S$ is much weaker than in the ReLU case. This relaxation is nontrivial, and the proof stems from a property of the derivatives of exponential functions. Consider the exponential function $\exp(w \cdot x)$ as a function of $w \in B(w^*, \delta)$, and denote it as $h(w)$,

$$h(w) = \exp(w \cdot x) = \mathrm{e}^{w_1 x_1 + \cdots + w_d x_d}, \quad w, \, x \in \mathbb{R}^d, \, d = d_x, \qquad (25)$$

where $w_i$ and $x_i \in \mathbb{R}$ are the components of $w$ and $x$, respectively. Calculating the partial derivatives of $h(w)$, we observe the following relations:

$$\frac{\partial^\alpha h}{\partial w^\alpha} := \frac{\partial^{|\alpha|} h}{\partial w_1^{\alpha_1} \cdots \partial w_d^{\alpha_d}} = x_1^{\alpha_1} \cdots x_d^{\alpha_d} h(w), \qquad (26)$$

where $\alpha = (\alpha_1, \ldots, \alpha_d) \in \mathbb{N}^d$ is the index vector representing the order of partial derivatives, and $|\alpha| := \alpha_1 + \cdots + \alpha_d$. This relationship allows us to link exponential FNNs to polynomials since any polynomial $P(x)$ can be represented in the following form:

$$P(x) = \exp(-w^* \cdot x) \left( \sum_{\alpha \in \Lambda} a_\alpha \frac{\partial^{|\alpha|} h}{\partial w^\alpha} \right) \Bigg|_{w=w^*}, \qquad (27)$$

where $a_\alpha$ are the coefficients of the polynomials, $\Lambda$ is a finite set of indices, and the partial derivatives can be approximated by finite differences, which are FNNs. For example, the first-order partial derivative $\frac{\partial h}{\partial w_1}\big|_{w=w^*} = x_1 h(w^*)$ can be approximated by the following difference with a small nonzero number $\lambda \in (0, \delta)$,

$$\frac{h(w^* + \lambda e_1) - h(w^*)}{\lambda} = \lambda^{-1}\exp((w^* + \lambda e_1)\cdot x) - \lambda^{-1}\exp(w^*\cdot x). \tag{28}$$

This is an exponential FNN with two neurons. Finally, employing the well-known Stone-Weierstrass Theorem, which states that any continuous function $f$ on compact domains can be approximated by polynomials, and combining the above relations between FNNs and polynomials, we can establish the UAP of exponential FNNs with weight constraints. When $y^{(i)} = f(x^{(i)})$ (referred to as meaningfully related), the conclusion still holds in standard ICL, provided that $\mathcal{V}_y$ satisfies certain conditions. A brief proof is provided in Theorem 15.

**Remark 9.** *When discussing density, one of the most immediate examples that comes to mind is the density of rational numbers in $\mathbb{R}$. How can we effectively enumerate rational numbers? The work by [54] introduces an elegant method for enumerating positive rational numbers, synthesizing ideas from [55] and [56]. It demonstrates the computational feasibility of enumeration through an effective algorithm. Thus, we assume that positional encodings can be implemented using computer algorithms, such as iterative functions. Furthermore, since positional encodings vary across different positions, they encapsulate semantic information concerning both position and order.*

## 5   Conclusion

In this paper, we establish a connection between FNNs and Transformers through ICL. By leveraging the UAP of FNNs, we demonstrate that the UAP of ICL holds when the context is selected from the entire vector space. When the context is drawn from a finite set, we explore the approximation power of VICL, showing that the UAP is achievable only when appropriate positional encodings are incorporated, highlighting their importance.

In our work, we consider Transformers with input sequences of arbitrary length, implying that the positional encoding $\mathcal{P}_x$ consists of a countably infinite set of elements. In Theorem 7, we assume a strong density condition, which is later relaxed in Theorem 8. However, in practical applications, input sequences are finite and are typically truncated for computational feasibility. This shift allows our conclusions to be interpreted through an approximation lens, where the objective is to approximate functions within a specified error margin, rather than achieving infinitesimal precision. Additionally, to achieve the UAP, it is insightful to compare the function approximation capabilities of our approach (outlined in Lemma 4) with the direct use of FNNs, particularly when the Transformer parameters are trainable.

It is important to note that this paper is limited to single-layer Transformers with APEs, and the main results (Theorem 7 and Theorem 8) focus on element-wise activations. Future research should extend these findings to multi-layer Transformers, general positional encodings (such as RPEs and RoPE), and softmax activations. For softmax Transformers, our analysis in Sections 2 and 3 highlighted their connection to Transformers with exponential activations. However, extending this connection to the scenario in Section 4 proves challenging and requires more sophisticated techniques.

Although this paper primarily addresses theoretical issues, we believe our results provide valuable insights for practitioners. In Remark 9, we observe that algorithms using function composition to enumerate dense numbers in $\mathbb{R}$ could inspire positional encodings via compositions of fixed functions, similar to RNN approaches. RNNs capture the sequential nature of information by integrating word order. However, existing research on RNNs has not explored the denseness of the sets formed by their hidden states, which we hope will inspire future experimental research. Lastly, our construction for Theorem 7 relies on the sparse partition assumption in equation (14), whose practical validity remains uncertain and requires future exploration.

In fact, Tack et al. [57], Hao et al. [58] on continuous CoTs and continuous states are closely related to our work – specifically, leveraging positional encoding to enable Transformers to achieve the UAP for functions whose domain is a finite set while the range covers the entire Euclidean space. Moreover, Xiao et al. [59] propose an approach for automatically adjusting prompts for function fitting, which is also related to our theoretical findings. Therefore, with further research, our theory holds practical significance.

## Acknowledgements

This work was partially supported by the National Key R&D Program of China under grants 2024YFF0505501 and the Fundamental Research Funds for the Central Universities. We thank the anonymous reviewers for their valuable comments and helpful suggestions. We gratefully acknowledge the Scholar Award from the Neural Information Processing Foundation.

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

# A Table of Notations

We present all our notations in Table 1 for easy reference.

Table 1: Table of Notations

| Notations | Explanations |
|---|---|
| $d_x$, $d_y$ | Dimensions of input and output. |
| $\mathcal{P}$ | Positional encoding. |
| $X, Y$ | Context without positional encoding. |
| $X_\mathcal{P}$, $Y_\mathcal{P}$ | Context with positional encoding $\mathcal{P}$. |
| $Z$ | Input without positional encoding. |
| $Z_\mathcal{P}$ | Input with positional encoding $\mathcal{P}$. |
| $\mathcal{V}$ | Vocabulary. |
| $\mathcal{V}_x$, $\mathcal{V}_y$ | Vocabulary of $x^{(i)}$ and $y^{(i)}$. |
| $\sigma$ | Activation function. |
| $\#$ | Cardinality of a set. |
| $\mathrm{N}^\sigma$, $\mathcal{N}^\sigma$ | One-hidden-layer FNN and its collection. |
| $\mathrm{T}^\sigma$, $\mathcal{T}^\sigma$ | Single-layer Transformer and its collection. |
| $\mathrm{N}^\sigma_*$, $\mathcal{N}^\sigma_*$ | One-hidden-layer FNN with a finite set of weights and its collection. |
| $\mathrm{T}^\sigma_*$, $\mathcal{T}^\sigma_*$ | Single-layer Transformer with vocabulary restrictions and its collection. |
| $\mathrm{T}^\sigma_{*,\mathcal{P}}$, $\mathcal{T}^\sigma_{*,\mathcal{P}}$ | Single-layer Transformer with positional encoding, vocabulary restrictions, and its collection. |
| $\|\cdot\|$ | The uniform norm of vectors, *i.e.*, a shorthand for $\|\cdot\|_\infty$. |
| $\tilde{x}$ | Append a one to the end of $x$, *i.e.*, $\tilde{x} = \begin{bmatrix} x \\ 1 \end{bmatrix}$. |

# B Proofs for Section 2

We provide detailed proofs of lemmas in Section 2. We will first directly prove Lemma 3 using Lemma 2. Next, by a similar method together with an additional technical refinement, we will establish Lemma 11. Finally, leveraging Lemma 11, we will prove Lemma 4.

## B.1 Proof of Lemma 3

**Lemma 3.** *Let $\sigma : \mathbb{R} \to \mathbb{R}$ be a non-polynomial, locally bounded, piecewise continuous activation function, and $\mathrm{T}^\sigma$ be a single-layer Transformer satisfying Assumption 1. For any one-hidden-layer network $\mathrm{N}^\sigma : \mathbb{R}^{d_x-1} \to \mathbb{R}^{d_y} \in \mathcal{N}^\sigma$ with $n$ hidden neurons, there exist matrices $X \in \mathbb{R}^{d_x \times n}$ and $Y \in \mathbb{R}^{d_y \times n}$ such that*

$$\mathrm{T}^\sigma\left(\tilde{x}; X, Y\right) = \mathrm{N}^\sigma(x), \quad \forall x \in \mathbb{R}^{d_x-1}. \tag{29}$$

*Proof.* The output of $\mathrm{T}^\sigma$ can be computed directly as

$$
\begin{aligned}
\mathrm{T}^\sigma(\tilde{x}, X, Y) &= \left(Z + \mathrm{Attn}^\sigma_{Q,K,V}(\tilde{x}, X, Y)\right)_{d_x+1:d_x+d_y, n+1} \\
&= \left(Z + VZM\sigma(Z^\top Q^\top KZ)\right)_{d_x+1:d_x+d_y, n+1} \\
&= \left(Z + \begin{bmatrix} DX+Ey & 0 \\ UY & 0 \end{bmatrix} \begin{bmatrix} \sigma(X^\top B^\top CX) & \sigma(X^\top B^\top C\tilde{x}) \\ \sigma(\tilde{x}^\top B^\top CX) & \sigma(\tilde{x}^\top B^\top C\tilde{x}) \end{bmatrix}\right)_{d_x+1:d_x+d_y, n+1} \\
&= UY\sigma(X^\top B^\top C\tilde{x}).
\end{aligned}
\tag{30}
$$

One can easily observe that the output closely resembles that of a single-layer FNN. Suppose $\mathrm{N}^\sigma(x) = A\sigma(Wx + b) : \mathbb{R}^{d_x-1} \to \mathbb{R}^{d_y}$ is an arbitrary single-layer FNN with $k$ hidden neurons, where $W \in \mathbb{R}^{k \times (d_x-1)}$, $A \in \mathbb{R}^{d_y \times k}$ and $b \in \mathbb{R}^k$. We construct the context by setting its length to $k$, *i.e.*, $X \in \mathbb{R}^{d_x \times k}$, $Y \in \mathbb{R}^{d_y \times k}$. A straightforward calculation shows that choosing

$$X = (C^\top B)^{-1} \begin{bmatrix} W & b \end{bmatrix}^\top, \quad Y = U^{-1}A, \tag{31}$$

suffices to guarantee $T^\sigma(\tilde{x}; X, Y) = N^\sigma(x)$ for all $x \in \mathbb{R}^{d_x - 1}$. $\qquad\qquad\square$

**Remark 10.** *It is worth noting that in the above proof, the matrix $F$ was set to zero in accordance with Assumption 1. However, we emphasize that this is not a strict requirement. In fact, one can accommodate an arbitrary $F$ by choosing $Y = U^{-1}(A - FX)$. The choice $F = 0$ is made purely for computational convenience under our current assumptions, which is also discussed in Appendix F.*

## B.2 Proof of the UAP of Softmax FNNs

Before proving Lemma 4, we demonstrate the UAP of softmax FNNs as a supporting lemma.

**Lemma 11** (UAP of Softmax FNNs). *For any continuous function $f : \mathbb{R}^{d_x} \to \mathbb{R}^{d_y}$ defined on a compact domain $\mathcal{K}$, and for any $\varepsilon > 0$, there exists a network $N^{\text{softmax}}(x) : \mathbb{R}^{d_x} \to \mathbb{R}^{d_y}$ satisfying*

$$\|N^{\text{softmax}}(x) - f(x)\| < \varepsilon, \quad \forall x \in \mathcal{K}. \tag{32}$$

*Proof.* We first build a bridge connecting softmax FNNs and target function $f$ using Lemma 2. We can construct a network

$$N^{\exp}(x) = A \exp(Wx + b) = \sum_{i=1}^{k} a_i e^{w_i \cdot x + b_i}, \tag{33}$$

with $k$ hidden neurons such that

$$\max_{x \in \mathcal{K}} \|N^{\exp}(x) - f(x)\| < \frac{\varepsilon}{2}, \tag{34}$$

where $a_i \in \mathbb{R}^{d_y}$, $w_i \in \mathbb{R}^{d_x}$, $b_i \in \mathbb{R}$. It therefore suffices to construct a softmax FNN $N^{\text{softmax}}(x)$ that approximates $N^{\exp}(x)$. This task amounts to eliminating the effect of the normalization in the softmax output.

Consider a softmax FNN

$$N^{\text{softmax}}(x) = A' \operatorname{softmax}\left(W'x + b'\right) = \frac{\displaystyle\sum_{i=1}^{k+1} a_i' e^{w_i' \cdot x + b_i'}}{\displaystyle\sum_{j=1}^{k+1} e^{w_j' \cdot x + b_j'}}, \tag{35}$$

with $k + 1$ hidden neurons, where $w_{k+1}' = b_{k+1}' = 0$, $b_i' = b_i'(\varepsilon)$ is sufficiently small such that

$$e^{w_i' \cdot x + b_i'} < \frac{\varepsilon}{2k\left(1 + \max_{x \in \mathcal{K}} \|N^{\exp}(x)\|\right)}, \quad \forall x \in \mathcal{K}, \ i = 1, 2, \cdots, k, \tag{36}$$

and $w_i' = w_i$ for $i = 1, 2, \cdots, k$. This arrangement ensures that, in the denominators of each term in Equation (35), the first $k$ entries are arbitrarily small, while the $(k + 1)$-th entry is exactly one. We then simply adjust $A'$ so that the numerators coincide with those in Equation (33), and this can be done by setting $a_i' = \begin{cases} a_i e^{b_i - b_i'}, & i = 1, 2, \cdots, k \\ 0, & i = k + 1 \end{cases}$. With the formal construction now complete, we present a more precise estimate of the approximation error as follows.

$$\|\mathrm{N}^{\mathrm{exp}}(x) - \mathrm{N}^{\mathrm{softmax}}(x)\| = \max_{x \in \mathcal{K}} \left\| \sum_{i=1}^{k} a_i \mathrm{e}^{w_i \cdot x + b_i} - \frac{\sum_{i=1}^{k+1} a'_i \mathrm{e}^{w'_i \cdot x + b'_i}}{\sum_{j=1}^{k+1} \mathrm{e}^{w'_j \cdot x + b'_j}} \right\|$$

$$= \max_{x \in \mathcal{K}} \left\| \sum_{i=1}^{k} a_i \mathrm{e}^{w_i \cdot x + b_i} - \frac{\sum_{i=1}^{k} a_i \mathrm{e}^{w_i \cdot x + b_i}}{\sum_{j=1}^{k} \mathrm{e}^{w'_j \cdot x + b'_j} + 1} \right\|$$

$$= \max_{x \in \mathcal{K}} \|\mathrm{N}^{\mathrm{exp}}(x)\| \cdot \max_{x \in \mathcal{K}} \left\| 1 - \frac{1}{\sum_{j=1}^{k} \mathrm{e}^{w'_j \cdot x + b'_j} + 1} \right\| \tag{37}$$

$$\leq \max_{x \in \mathcal{K}} \|\mathrm{N}^{\mathrm{exp}}(x)\| \cdot \max_{x \in \mathcal{K}} \left\| \sum_{j=1}^{k} \mathrm{e}^{w'_j \cdot x + b'_j} \right\|$$

$$\leq \frac{\varepsilon}{2}.$$

This leads to the conclusion that $\|\mathrm{N}^{\mathrm{softmax}}(x) - f(x)\| < \varepsilon$ for all $x \in \mathcal{K}$, thus completing the proof. $\qquad\square$

### B.3 Proof of Lemma 4

**Lemma 4.** *Let $\sigma : \mathbb{R} \to \mathbb{R}$ be a non-polynomial, locally bounded, piecewise continuous activation function or softmax function, and $\mathrm{T}^\sigma$ be a single-layer Transformer satisfying Assumption 1, and $\mathcal{K}$ be a compact domain in $\mathbb{R}^{d_x - 1}$. Then for any continuous function $f : \mathcal{K} \to \mathbb{R}^{d_y}$ and any $\varepsilon > 0$, there exist matrices $X \in \mathbb{R}^{d_x \times n}$ and $Y \in \mathbb{R}^{d_y \times n}$ such that*

$$\left\| \mathrm{T}^\sigma(\tilde{x}; X, Y) - f(x) \right\| < \varepsilon, \quad \forall x \in \mathcal{K}. \tag{38}$$

*Proof.* For element-wise activation case, with the help of Lemma 2 and Lemma 3, the conclusion follows trivially.

Then we handle the softmax case. Similarly, for any $\varepsilon > 0$, we can construct a softmax FNN $\mathrm{N}^{\mathrm{softmax}}(x)$ with $k$ hidden neurons, using Lemma 11 such that

$$\max_{x \in \mathcal{K}} \|\mathrm{N}^{\mathrm{softmax}}(x) - f(x)\| < \frac{\varepsilon}{2}. \tag{39}$$

We need to approximate this softmax FNN with a softmax Transformer. The computation proceeds as follows:

$$\mathrm{T}^{\mathrm{softmax}}(\tilde{x}, X, Y)$$
$$= \left( Z + \begin{bmatrix} DX + EY & 0 \\ UY & 0 \end{bmatrix} \mathrm{softmax} \left( \begin{bmatrix} X^\top B^\top C X & X^\top B^\top C \tilde{x} \\ \tilde{x}^\top B^\top C X & \tilde{x}^\top B^\top C \tilde{x} \end{bmatrix} \right) \right)_{d_x + 1 : d_x + d_y, n+1} \tag{40}$$
$$= UY \left( \mathrm{softmax} \left( \begin{bmatrix} X^\top B^\top C \tilde{x} \\ \tilde{x}^\top B^\top C \tilde{x} \end{bmatrix} \right) \right)_{1:n}.$$

By comparing the output with that of the exponential FNN, we find that there is an additional bounded positive term $t(x) := \exp\left(\tilde{x}^\top B^\top C \tilde{x}\right)$ arising from the normalization process.

Choose the length of the context $n = k + 1$ and matrices $X, Y$ such that

$$X^\top B^\top C = \begin{bmatrix} W & b + s\mathbf{1} \\ 0 & s \end{bmatrix}, \ UY = [A \quad 0], \tag{41}$$

where $\mathbf{1}$ is all-ones vector and $s$ is big enough, making

$$e^{\tilde{x}^\top B^\top C\tilde{x}-s} < \frac{\varepsilon}{2\left(1+\max_{x\in\mathcal{K}}\|\mathrm{N}^{\mathrm{softmax}}(x)\|\right)}, \quad \forall x\in\mathcal{K}. \tag{42}$$

Then $X^\top B^\top C\tilde{x} = \begin{bmatrix} W & b+s\mathbf{1} \\ 0 & s \end{bmatrix} \begin{bmatrix} x \\ 1 \end{bmatrix} = \begin{bmatrix} Wx+b+s\mathbf{1} \\ s \end{bmatrix}$, and we can compute the explicit form of $\mathrm{T}^{\mathrm{softmax}}(\tilde{x};X,Y)$ as:

$$
\begin{aligned}
\mathrm{T}^{\mathrm{softmax}}(\tilde{x};X,Y) &= \frac{\displaystyle\sum_{i=1}^{k} a_i \exp(w_i\cdot x+b_i+s)}{\displaystyle\sum_{j=1}^{k} \exp(w_j\cdot x+b_j+s)+\exp(s)+\exp(\tilde{x}^\top B^\top C\tilde{x})} \\
&= \frac{\displaystyle\sum_{i=1}^{k} a_i \exp(w_i\cdot x+b_i)}{\displaystyle\sum_{j=1}^{k} \exp(w_j\cdot x+b_j)+1+\exp(\tilde{x}^\top B^\top C\tilde{x}-s)}.
\end{aligned}
\tag{43}
$$

We estimate the upper bound of the distance between $\mathrm{N}^{\mathrm{softmax}}$ and $\mathrm{T}^{\mathrm{softmax}}$, that is

$$
\begin{aligned}
&\max_{x\in\mathcal{K}} \|\mathrm{N}^{\mathrm{softmax}}(x) - \mathrm{T}^{\mathrm{softmax}}(\tilde{x};X,T)\| \\
&= \max_{x\in\mathcal{K}} \left\| \frac{\displaystyle\sum_{i=1}^{k} a_i\exp(w_i\cdot x+b_i)}{\displaystyle\sum_{j=1}^{k}\exp(w_j\cdot x+b_j)+1} - \frac{\displaystyle\sum_{i=1}^{k} a_i\exp(w_i\cdot x+b_i)}{\displaystyle\sum_{j=1}^{k}\exp(w_j\cdot x+b_j)+1+\exp(\tilde{x}^\top B^\top C\tilde{x}-s)} \right\| \\
&= \max_{x\in\mathcal{K}} \|\mathrm{N}^{\mathrm{softmax}}(x)\| \cdot \max_{x\in\mathcal{K}} \left\| 1 - \frac{\displaystyle\sum_{j=1}^{k}\exp(w_j\cdot x+b_j)+1}{\displaystyle\sum_{j=1}^{k}\exp(w_j\cdot x+b_j)+1+\exp(\tilde{x}^\top B^\top C\tilde{x}-s)} \right\| \\
&= \max_{x\in\mathcal{K}} \|\mathrm{N}^{\mathrm{softmax}}(x)\| \cdot \max_{x\in\mathcal{K}} \left\| \frac{\exp(\tilde{x}^\top B^\top C\tilde{x}-s)}{\displaystyle\sum_{j=1}^{k}\exp(w_j\cdot x+b_j)+1+\exp(\tilde{x}^\top B^\top C\tilde{x}-s)} \right\| \\
&\leq \max_{x\in\mathcal{K}} \|\mathrm{N}^{\mathrm{softmax}}(x)\| \cdot \max_{x\in\mathcal{K}} \left\| \exp(\tilde{x}^\top B^\top C\tilde{x}-s) \right\| \\
&\leq \frac{\varepsilon}{2}.
\end{aligned}
\tag{44}
$$

As a consequence, we have $\left\| \mathrm{T}^\sigma\left(\tilde{x};X,Y\right) - f(x)\right\| < \varepsilon$ for all $x\in\mathcal{K}$, which finishes the proof. $\quad\square$

## C  Proofs for Section 3

In this appendix, we provide detailed proofs of Proposition 12, Lemma 5, and Theorem 6 presented in Section 3. We will first use induction to prove Proposition 12, and then employ this proposition together with a proof by contradiction to establish Lemma 5 and Theorem 6.

**Proposition 12.** *The scalar function $h_k(x) = \sum_{i=1}^{k} a_i e^{b_i x}$, with $a_i$, $b_i$, $x\in\mathbb{R}$, where the exponents $b_i$ are pairwise distinct, and at least one $a_i$ is nonzero, has at most $k-1$ zero points.*

*Proof.* We prove this statement by induction. When $k=1$ or 2, this can be proven easily. We suppose $h_N(x)$ has at most $N-1$ zero points. Now consider the case $k=N+1$. Let $h_{N+1}(x) =$

$\sum_{i=1}^{N+1} a_i \mathrm{e}^{b_i x}$. Without loss of generality, assume that $a_{N+1} \neq 0$. Thus, we can rewrite $h_{N+1}(x)$ as

$$h_{N+1}(x) = a_{N+1}\mathrm{e}^{b_{N+1}x}\Big(1 + \sum_{i=1}^{N} \frac{a_i}{a_{N+1}}\mathrm{e}^{(b_i - b_{N+1})x}\Big) := a_{N+1}\mathrm{e}^{b_{N+1}x}g(x).$$

Then we process by contradiction. Suppose $h_{N+1}(x)$ has more than $N$ zero points, which implies $g(x)$ has more than $N$ zero points. Then, according to Rolle's Theorem, $g'(x)$ must have more than $N-1$ zero points, which contradicts our assumption. Thus, $h_{N+1}$ have at most $N$ zero points, and the proof is complete. $\qquad\square$

**Lemma 5.** *The function class $\mathcal{N}_*^\sigma$, with a non-polynomial, locally bounded, piecewise continuous element-wise activation function or softmax activation function $\sigma$, cannot achieve the UAP. Specifically, there exist a compact domain $\mathcal{K} \subset \mathbb{R}^{d_x}$, a continuous function $f : \mathcal{K} \to \mathbb{R}^{d_y}$, and $\varepsilon_0 > 0$ such that*

$$\max_{x \in \mathcal{K}} \big\| f(x) - \mathrm{N}_*^\sigma(\tilde{x}) \big\| \geq \varepsilon_0, \quad \forall\, \mathrm{N}_*^\sigma \in \mathcal{N}_*^\sigma. \tag{45}$$

*Proof.* For any element-wise activation $\sigma$, $\mathrm{span}\{\mathcal{N}^\sigma\}$ forms a finite-dimensional function space. $\mathrm{span}\{\mathcal{N}^\sigma\}$ is closed under the uniform norm as established by Theorem 2.1 from [52] and Corollary C.4 from [53]. This implies that the set of functions approximable by $\mathrm{span}\{\mathcal{N}^\sigma\}$ is precisely the set of functions within $\mathrm{span}\{\mathcal{N}^\sigma\}$. Consequently, any function not in $\mathrm{span}\{\mathcal{N}^\sigma\}$ cannot be arbitrarily approximated, meaning that the UAP cannot be achieved.

Then we prove the softmax case. First, we simplify the problem to facilitate the construction of a function that cannot be approximated. We observe that it suffices to prove the UAP fails when the first input coordinate ranges over $[0, 1]$ and all other coordinates are held fixed. Indeed, we can find the compact set $K \subset \mathbb{R}^{d_x}$ containing $\prod_{i=1}^{d_x}[l_i, r_i]$. If we can show that $\mathcal{N}^{\mathrm{softmax}}$ does not achieve the UAP on $[l_1, r_1] \times \prod_{i=2}^{d_x}\{l_i\}$, then, by applying a suitable affine change of variables, it follows that UAP also fails on $[0, 1] \times \prod_{i=2}^{d_x}\{l_i\}$. Consider a continuous target function

$$f : [0, 1] \times \prod_{i=2}^{d_x}\{l_i\} \to \mathbb{R}, \ (x_1, x_2, \cdots, x_{d_x}) \mapsto f_1(x_1). \tag{46}$$

The reason why we consider such a target function is that every vector-valued function $f(x_1, \cdots, x_{d_x})$ can be represented as $f(x_1, \cdots, x_{d_x}) = \big(f_1(x_1, \cdots, x_{d_x}), \cdots, f_{d_y}(x_1, \cdots, x_{d_x})\big)$. If the UAP fails for $f$, it must fail for at least one of its scalar components. Hence it suffices to consider the one-dimensional (scalar) case. Moreover, since the values of $x_2, \cdots, x_{d_x}$ are fixed, the above reduction to a single-variable scalar function is justified. We only need to demonstrate that there exists at least one such function that cannot be approximated arbitrarily well by any $\mathrm{N}_*^{\mathrm{softmax}} \in \mathcal{N}_*^{\mathrm{softmax}}$.

Then we will use Proposition 12 to complete the remainder of this proof. Before that, we need to rewrite the form of the output of $\mathrm{N}^{\mathrm{softmax}}$, which is

$$\mathrm{N}_*^{\mathrm{softmax}}(x) = \frac{\sum_{i=1}^{k} a_i \mathrm{e}^{w_i \cdot x + b_i}}{\sum_{j=1}^{k} \mathrm{e}^{w_j \cdot x + b_j}}, \tag{47}$$

where $(a_i, w_i, b_i) \in \mathcal{A} \times \mathcal{W} \times \mathcal{B}$ is a finite set and $k$ is the number of hidden neurons. Consequently, the set $\mathcal{W} \times \mathcal{B}$ is finite, and we denote it as $N := \#(\mathcal{W} \times \mathcal{B})$. By regrouping identical terms in the numerator, we can rewrite the equation as

$$\mathrm{N}_*^{\mathrm{softmax}}(x) = \frac{\sum_{i=1}^{N} \tilde{a}_i \mathrm{e}^{w_i \cdot x + b_i}}{\sum_{j=1}^{k} \mathrm{e}^{w_j \cdot x + b_j}}. \tag{48}$$

It is important to note that this transformation applies to any $\mathrm{N}_*^{\mathrm{softmax}} \in \mathcal{N}_*^{\mathrm{softmax}}$, ensuring that the number of summation terms in the numerator remains strictly bounded by $N$.

Finally, we construct a function which cannot be approximated by such softmax networks. Assume a continuous target function

$$g : [0,1] \times \prod_{i=2}^{d_x} \{l_i\} \to \mathbb{R}, \ (x_1, x_2, \cdots, x_{d_x}) \mapsto \cos\big((N+1)\pi x_1\big), \tag{49}$$

which has $(N+1)$ zero points. If $\mathcal{N}_*^{\text{softmax}}$ achieves the UAP, we assume that $\mathrm{N}_*^{\text{softmax}} \in \mathcal{N}_*^{\text{softmax}}$ which satisfies $\|\mathrm{N}_*^{\text{softmax}} - g\| \le \varepsilon < \frac{1}{10}$. We denote $z_i = \frac{i}{N+1}$ for $i = 0, 1, \cdots, N+1$. It is easy to verify that $g(z_i) = 1$ if $i$ is even, and $g(z_i) = -1$ if $i$ is odd, which means $\mathrm{N}_*^{\text{softmax}}(z_i) > 0.9$ for even $i$ and $\mathrm{N}_*^{\text{softmax}}(z_i) < -0.9$ for odd $i$. According to the Intermediate Value Theorem, $\mathrm{N}_*^{\text{softmax}}$ has at least $N+1$ zero points, which contradicts Proposition 12. And we finish our proof. □

We will use Figure 1 to provide readers with an intuitive illustration of why a class of functions whose number of zeros is bounded cannot achieve universal approximation.

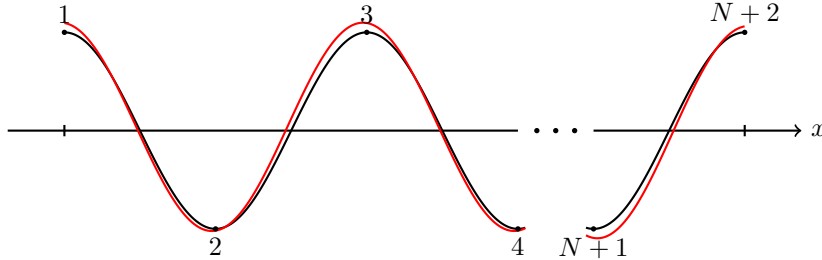

Figure 1: An illustration of non-approximability. The black curve represents the target function, which has $N+1$ zero points. The red curve represents a sum of exponentials, which has no more than $N$ zero points. If the UAP holds, then the red curve must pass near the $N+2$ marked extrema in the figure. By the Intermediate Value Theorem, the function represented by the red curve would then have $N+1$ zeros, which contradicts its intrinsic properties.

### C.1 Proof of Theorem 6

**Theorem 6.** *The function class $\mathcal{T}_*^\sigma$, with a non-polynomial, locally bounded, piecewise continuous element-wise activation function or softmax activation function $\sigma$ and every $\mathrm{T}^\sigma \in \mathcal{T}_*^\sigma$ satisfies Assumption 1, cannot achieve the UAP. Specifically, there exist a compact domain $\mathcal{K} \subset \mathbb{R}^{d_x}$, a continuous function $f : \mathcal{K} \to \mathbb{R}^{d_y}$, and $\varepsilon_0 > 0$ such that*

$$\max_{x \in \mathcal{K}} \big\| f(x) - \mathrm{T}_*^\sigma(\tilde{x}) \big\| \ge \varepsilon_0, \quad \forall \, \mathrm{T}_*^\sigma \in \mathcal{T}_*^\sigma. \tag{50}$$

*Proof.* For cases of element-wise activation, since $\mathrm{T}_*^\sigma$ has a similar structure to $\mathrm{N}_*^\sigma$, we find that $\mathrm{span}\{\mathrm{T}_*^\sigma\}$ is also a finite-dimensional function space. Hence, the same argument from Lemma 5 can be applied here to complete the proof.

Then we prove the softmax case. Recall equation (40), the output of $\mathrm{T}_*^{\text{softmax}}(\tilde{x}; X, Y)$ can be viewed as

$$\mathrm{T}_*^{\text{softmax}}(\tilde{x}; X, Y) = \frac{\displaystyle\sum_{i=1}^{n} a_i \mathrm{e}^{w_i \cdot x + b_i}}{\displaystyle\sum_{j=1}^{n} \mathrm{e}^{w_j \cdot x + b_j} + \mathrm{e}^{\tilde{x}^\top B^\top C \tilde{x}}}, \tag{51}$$

where $n$ denotes the context length and $a_i \in \mathcal{A}$, $w_i \in \mathcal{W}$, $b_i \in \mathcal{B}$ for some finite sets $\mathcal{A}$, $\mathcal{W}$, $\mathcal{B}$. This allows us to apply the same approach as in the proof of Lemma 5, which leads to the conclusion that $\mathcal{T}_*^\sigma$ cannot achieve the UAP. □

# D    Kronecker Approximation Theorem

To facilitate our constructive proof, we introduce the Kronecker Approximation Theorem as an auxiliary tool to support the main theorem.

**Lemma 13** (Kronecker Approximation Theorem [60])**.** *Given real $n$-tuples $\alpha^{(i)} = (\alpha_1^{(i)}, \alpha_2^{(i)}, \cdots, \alpha_n^{(i)}) \in \mathbb{R}^n$ for $i = 1, \cdots, m$ and $\beta = (\beta_1, \beta_2, \cdots, \beta_n) \in \mathbb{R}^n$, the following condition holds: for any $\varepsilon > 0$, there exist $q_i, l_j \in \mathbb{Z}$ such that*

$$\left\| \beta_j - \sum_{i=1}^{m} q_i \alpha_j^{(i)} + l_j \right\| < \varepsilon, \quad j = 1, \cdots, n, \tag{52}$$

*if and only if for any $r_1, \cdots, r_n \in \mathbb{Z}$, $i = 1, \cdots, m$ with*

$$\sum_{j=1}^{n} \alpha_j^{(i)} r_j \in \mathbb{Z}, \quad i = 1, \cdots, m, \tag{53}$$

*the number $\sum\limits_{j=1}^{n} \beta_j r_j$ is also an integer. In the case of $m = 1$ and $n = 1$, for any $\alpha$, $\beta \in \mathbb{R}$ with $\alpha$ irrational and $\varepsilon > 0$, there exist integers $l$ and $q$ with $q > 0$ such that $|\beta - q\alpha + l| < \varepsilon$.*

Lemma 13 indicates that if the condition in equation (53) is satisfied only when all $r_i$ are zeros, then the set $\{Mq + l \mid q \in \mathbb{Z}^m, \ l \in \mathbb{Z}^n\}$ is dense in $\mathbb{R}^n$, where the matrix $M \in \mathbb{R}^{n \times m}$ is assembled with vectors $\alpha^{(i)}$, i.e., $M = [\alpha^{(1)}, \alpha^{(2)}, \cdots, \alpha^{(m)}]$. In the case of $m = 1$ and $n = 1$, let $\alpha = \sqrt{2}$, then Lemma 13 implies that the set $\{q\sqrt{2} \pm l \mid l \in \mathbb{N}^+, \ q \in \mathbb{N}^+\}$ is dense in $\mathbb{R}$. We will build upon this result to prove one of the main theorems in this article.

# E    Proofs for Section 4

In this appendix, we lay the groundwork for the proof of Theorem 7 by first introducing Lemma 14. We then present Theorem 7 and provide its complete proof, demonstrating that $\mathcal{T}_{*,\mathcal{P}}^{\sigma}$ can realize the UAP. To facilitate understanding of Theorem 7, we provide a simple illustrative example. While the theorem assumes dense positional encodings, we relax this condition under specific activation functions, as formalized in Lemma 16 and Theorem 8.

## E.1    Lemma 14

**Lemma 14.** *For a network with a fixed width and a continuous activation function, it is possible to apply slight perturbations within an arbitrarily small error margin. For any network $N_1^{\sigma}(x)$ defined on a compact set $\mathcal{K} \subset \mathbb{R}^{d_x}$, with parameters $A \in \mathbb{R}^{d_y \times k}, W \in \mathbb{R}^{k \times d_x}, b \in \mathbb{R}^{k \times 1}$, there exists $M > 0, M_1 > 0$ ($\|x\| < M$ and $\|a_i\| < M_1, i = 1, \cdots, k$ ), and for any $\varepsilon > 0$, there exists $0 < \delta < \frac{\varepsilon}{2 M_1 k}$ and a perturbed network $N_2^{\sigma}(x)$ with parameters $\tilde{A} \in \mathbb{R}^{d_y \times k}, \tilde{W} \in \mathbb{R}^{k \times d_x}, \tilde{b} \in \mathbb{R}^{k \times 1}$ ( $\left\| \sigma(\tilde{w}_i \cdot x + \tilde{b}_i) \right\| < M_1, i = 1, \cdots, k$), such that if $\max\{ \|a_i - \tilde{a}_i\|, M\|w_i - \tilde{w}_i\| + \|b - \tilde{b}\| \mid i = 1, \cdots, k\} < \delta$, then*

$$\|N_1^{\sigma}(x) - N_2^{\sigma}(x)\| < \varepsilon, \quad \forall x \in \mathcal{K}, \tag{54}$$

*where $a_i, \tilde{a}_i$ are the $i$-th column vectors of $A, \tilde{A}$, respectively, $w_i, \tilde{w}_i$ are the $i$-th row vectors of $W, \tilde{W}$, and $b_i, \tilde{b}_i$ are the $i$-th components of $b, \tilde{b}$, respectively, for any $i = 1, \cdots, k$.*

*Proof.* We have $N_1^{\sigma}(x) = \sum\limits_{i=1}^{k} a_i \sigma(w_i \cdot x + b_i)$, where $a_i \in \mathbb{R}^{d_y}, w_i \in \mathbb{R}^{d_x}, b_i \in \mathbb{R}$, and $\tilde{N}_2^{\sigma}(x) = \sum\limits_{i=1}^{k} \tilde{a}_i \sigma(\tilde{w}_i \cdot x + \tilde{b}_i)$, where $\tilde{a}_j \in \mathbb{R}^{d_y}, \tilde{w}_i \in \mathbb{R}^{d_x}, \tilde{b}_i \in \mathbb{R}$. For any $x \in \mathcal{K}, \|x\| < M$. There exists a constant $M_1 > 0$ such that for any $i = 1, \cdots, k$, the following inequalities hold: $\|a_i\| < M_1$ and $\left\| \sigma(\tilde{w}_i \cdot x + \tilde{b}_i) \right\| < M_1$.

Due to the continuity of the activation function, for any $\varepsilon > 0$, there exists $0 < \delta < \frac{\varepsilon}{2M_1 k}$, such that if $\|w_i \cdot x + b_i - (\tilde{w}_i \cdot x + \tilde{b}_i)\| \le \|w_i - \tilde{w}_i\| \|x\| + \|b_i - \tilde{b}_i\| < M\|w_i - \tilde{w}_i\| + \|b - \tilde{b}\| < \delta$, then $\|\sigma(w_i \cdot x + b_i) - \sigma(\tilde{w}_i \cdot x + \tilde{b}_i)\| < \frac{\varepsilon}{2M_1 k}$, and $\|a_i - \tilde{a}_i\| < \delta$, for any $i = 1, \cdots, k$.

Combining all these inequalities, we can further derive:

$$
\begin{aligned}
\|N_1^\sigma(x) - N_2^\sigma(x)\| &\left\| \sum_{i=1}^k a_i \sigma(w_i \cdot x + b_i) - \sum_{i=1}^k \tilde{a}_i \sigma(\tilde{w}_i \cdot x + \tilde{b}_i) \right\| \\
&\le \left\| \sum_{i=1}^k a_i \sigma(w_i \cdot x + b_i) - \sum_{i=1}^k a_i \sigma(\tilde{w}_i \cdot x + \tilde{b}_i) \right\| + \left\| \sum_{i=1}^k a_i \sigma(\tilde{w}_i \cdot x + \tilde{b}_i) - \sum_{i=1}^k \tilde{a}_i \sigma(\tilde{w}_i \cdot x + \tilde{b}_i) \right\| \\
&\le \max_i \|a_i\| \left\| \sum_{i=1}^k \sigma(w_i \cdot x + b_i) - \sum_{i=1}^k \sigma(\tilde{w}_i \cdot x + \tilde{b}_i) \right\| + \max_i \left\| \sigma(\tilde{w}_i \cdot x + \tilde{b}_i) \right\| \left\| \sum_{i=1}^k a_i - \sum_{i=1}^k \tilde{a}_i \right\| \cdot \\
&\le \max_i \|a_i\| \sum_{i=1}^k \left\| \sigma(w_i \cdot x + b_i) - \sigma(\tilde{w}_i \cdot x + \tilde{b}_i) \right\| + \max_i \left\| \sigma(\tilde{w}_i \cdot x + \tilde{b}_i) \right\| \sum_{i=1}^k \|a_i - \tilde{a}_i\| \\
&< M_1 k \frac{\varepsilon}{2M_1 k} + M_1 k \frac{\varepsilon}{2M_1 k} = \varepsilon.
\end{aligned}
$$

$$(55)$$

The proof is complete. $\qquad\square$

### E.2   Proof of Theorem 7

**Theorem 7.** *Let $\mathcal{T}_{*,\mathcal{P}}^\sigma$ be the class of functions $T_{*,\mathcal{P}}^\sigma$ satisfying Assumption 1, with a non-polynomial, locally bounded, piecewise continuous element-wise activation function $\sigma$, the subscript refers the finite vocabulary $\mathcal{V} = \mathcal{V}_x \times \mathcal{V}_y$, $\mathcal{P} = \mathcal{P}_x \times \mathcal{P}_y$ represents the positional encoding map, and denote a set $S$ as:*

$$S := \mathcal{V}_x + \mathcal{P}_x = \left\{ x_i + \mathcal{P}_x^{(j)} \,\middle|\, x_i \in \mathcal{V}_x,\ i,\ j \in \mathbb{N}^+ \right\}. \tag{56}$$

*If $S$ is dense in $\mathbb{R}^{d_x}$, $\{1,\ -1,\ \sqrt{2},\ 0\}^{d_y} \subset \mathcal{V}_y$ and $\mathcal{P}_y = 0$, then $\mathcal{T}_{*,\mathcal{P}}^\sigma$ can achieve the UAP. More specifically, given a network $T_{*,\mathcal{P}}^\sigma$, then for any continuous function $f : \mathbb{R}^{d_x-1} \to \mathbb{R}^{d_y}$ defined on a compact domain $\mathcal{K}$ and $\varepsilon > 0$, there always exist $X \in \mathbb{R}^{d_x \times n}$ and $Y \in \mathbb{R}^{d_y \times n}$ from the vocabulary $\mathcal{V}$, i.e., $x^{(i)} \in \mathcal{V}_x, y^{(i)} \in \mathcal{V}_y$, with some length $n \in \mathbb{N}^+$ such that*

$$\left\| T_{*,\mathcal{P}}^\sigma (\tilde{x}; X, Y) - f(x) \right\| < \varepsilon, \quad \forall x \in \mathcal{K}. \tag{57}$$

*Proof.* Our conclusion holds for all element-wise continuous activation functions in $\mathcal{T}_{*,\mathcal{P}}^\sigma$. We now assume $d_y = 1$ for simplicity, and the case $d_y \neq 1$ will be considered later.

We are reformulating the problem. Using Lemma 3, we have

$$T_{*,\mathcal{P}}^\sigma (\tilde{x}; X, Y) = U Y_\mathcal{P}\, \sigma \left( (X + \mathcal{P})^\top B^\top C \tilde{x} \right) = U Y_\mathcal{P}\, \sigma \left( X_\mathcal{P}^\top B^\top C \tilde{x} \right). \tag{58}$$

Since $\mathcal{P}_y = 0$, it follows that $Y_\mathcal{P} = Y$. For any continuous function $f : \mathbb{R}^{d_x-1} \to \mathbb{R}^{d_y}$ defined on a compact domain $\mathcal{K}$ and for any $\varepsilon > 0$, we aim to show that there exists $T_{*,\mathcal{P}}^\sigma \in \mathcal{T}_{*,\mathcal{P}}^\sigma$ such that:

$$
\begin{aligned}
&\left\| T_{*,\mathcal{P}}^\sigma \left( \begin{bmatrix} x \\ 1 \end{bmatrix}; X, Y \right) - U f(x) \right\| < \|U\| \varepsilon, \quad \forall x \in \mathcal{K}, \\
&\Leftrightarrow \left\| Y \sigma \left( X_\mathcal{P}^\top B^\top C \tilde{x} \right) - f(x) \right\| < \varepsilon, \quad \forall x \in \mathcal{K}.
\end{aligned}
\tag{59}
$$

In the main text, for illustrative purposes, we consider the special case where $U$ is the identity matrix to simplify the exposition. In the present analysis, we dispense with this assumption. We already have the Lemma 2 ensuring the existence of a one-hidden-layer network $N^\sigma$ (with activation function

$\sigma$ satisfying the required conditions) that approximates $f(x)$. Our proof is divided into four steps, serving as a bridge built upon the Lemma 2:

$$Y \sigma \left(X_{\mathcal{P}}^{\top} B^{\top} C \tilde{x}\right) \xrightarrow{\text{Lemma 2}} N_{*}^{\sigma}(x) \xrightarrow{\text{step (3)}} N'(x) \xrightarrow{\text{step (2)}} N^{\sigma}(x) \xrightarrow{\text{step (1)}} f(x). \tag{60}$$

We present the specific details at each step.

**Step (1): Approximating $f(x)$ Using $N^{\sigma}(x)$.**  Supported by Lemma 2, there exists a neural network $N^{\sigma}(x) = A \sigma(Wx+b) = \sum_{i=1}^{k} a_i \sigma(w_i \cdot x + b_i) \in \mathcal{N}^{\sigma}$, with parameters $k \in \mathbb{N}^{+}$, $A \in \mathbb{R}^{d_y \times k}$, $b \in \mathbb{R}^k$, and $W \in \mathbb{R}^{k \times (d_x - 1)}$,

$$\|A \sigma(Wx + b) - f(x)\| < \frac{\varepsilon}{3}, \quad \forall x \in \mathcal{K}. \tag{61}$$

**Step (2): Approximating $N^{\sigma}(x)$ Using $N'(x)$.**  Using Lemma 13 and Lemma 14, a neural network $N^{\sigma}(x) = \sum_{i=1}^{k} a_i \sigma(w_i \cdot x + b_i) \in \mathcal{N}^{\sigma}$ can be perturbed into $N'(x) = \sum_{i=1}^{k} (q\sqrt{2} \pm l)_i \, \sigma(\tilde{w}_i \cdot x + \tilde{b}_i)$ (with $q_i \in \mathbb{N}^{+}$ and $l_i \in \mathbb{N}^{+}, i = 1, \cdots, k$), such that for any $\varepsilon > 0$, there exists $0 < \delta < \frac{\varepsilon}{6M_1 k}$ satisfying:

$$\max\{\|a_i - (q\sqrt{2} \pm l)_i\|, \, M\|w_i - \tilde{w}_i\| + \|b - \tilde{b}\| \mid i = 1, \cdots, k\} < \delta, \tag{62}$$

ensuring:

$$\|N^{\sigma}(x) - N'(x)\| = \left\| \sum_{i=1}^{k} a_i \, \sigma(w_i \cdot x + b_i) - \sum_{i=1}^{k} (q\sqrt{2} \pm l)_i \, \sigma(\tilde{w}_i \cdot x + \tilde{b}_i) \right\| < \frac{\varepsilon}{3}, \quad \forall x \in \mathcal{K}. \tag{63}$$

**Step (3): Approximating $N'(x)$ Using $N_{*}^{\sigma}(x)$.**  Next, we show that $N_{*}^{\sigma}(x) = \sum_{i=1}^{n} y^{(i)} \sigma(\tilde{R}_i \cdot \tilde{x}) \in \mathcal{N}_{*}^{\sigma}$ can approximate $N'(x) = \sum_{i=1}^{k} (q\sqrt{2} \pm l)_i \, \sigma(\tilde{w}_i \cdot \tilde{x})$. As a demonstration, we approximate a single term $(q\sqrt{2} \pm l)_1 \, \sigma(\tilde{w}_1 \cdot \tilde{x})$. Since the positional encoding is fixed, *i.e.*, $\mathcal{V}_x + \mathcal{P}^{(1)}$ is a finite set, one of two cases must occur:

1. *Valid Position:* If there exists $x^{(1)} \in \mathcal{V}_x$ where $(x^{(1)} + \mathcal{P}^{(1)})^{\top} B^{\top} C \approx \tilde{w}_1$;

2. *Invalid Position:* Set $y^{(1)} = 0$ to nullify contribution.

Since $S$ is dense in $\mathbb{R}^{d_x}$ and $B^{\top} C$ is non-singular, the set $G := \{\tilde{R} \mid \tilde{R} = X_{\mathcal{P}}^{\top} B^{\top} C, X_{\mathcal{P}} \subset 2^S\}$ remains dense. Let $K_1$ denote the set of indices corresponding to all "valid" positions for $\tilde{w}_1$. Since $y^{(i)} \in \{1, -1, \sqrt{2}, 0\}$, we require $q_1 + l_1$ elements from $G$ that approximate $\tilde{w}_1$, such that

$$\begin{aligned} & \left\| \sum_{j \in K_1} y^{(j)} \sigma(\tilde{R}_j \cdot \tilde{x}) - (q\sqrt{2} \pm l)_1 \, \sigma(\tilde{w}_1 \cdot \tilde{x}) \right\| \\ & = \left\| \sqrt{2} \sum_{j \in Q_1} \sigma(\tilde{R}_j \cdot \tilde{x}) \pm \sum_{j \in L_1} \sigma(\tilde{R}_j \cdot \tilde{x}) - (q\sqrt{2} \pm l)_1 \, \sigma(\tilde{w}_1 \cdot \tilde{x}) \right\| \\ & < \frac{\varepsilon}{3k}, \quad \forall x \in \mathcal{K}. \end{aligned} \tag{64}$$

Here, $\#(K_1) = q_1 + l_1$ and $K_1 = Q_1 \bigcup L_1$, where $Q_1, L_1$ are disjoint subsets of positive integer indices satisfying $\#(Q_1) = q_1$ and $\#(L_1) = l_1$. For this construction, we assign $y^{(j)} = \sqrt{2}$ for $j \in Q_1$ and $y^{(j)} = \pm 1$ for $j \in L_1$. For $j \in \{1, 2, 3, \cdots, \max_i\{i \in K_1\}\} \backslash K_1$, *i.e.*, for the *Invalid Position*, we set $y^{(j)} = 0$.

The multi-term approximation employs parallel construction via disjoint node subsets $K_i = Q_i \cup L_i$, where $Q_i$ ($q_i$ nodes) and $L_i$ ($l_i$ nodes) implement $\sqrt{2}$ and $\pm 1$ coefficients respectively. For $j \notin \bigcup_{l=1}^{k} K_l$, we set $y^{(j)} = 0$. Each term achieves:

$$\left\| \sum_{j \in K_i} y^{(j)} \sigma(\tilde{R}_j \cdot \tilde{x}) - (q\sqrt{2} \pm l)_i \sigma(\tilde{w}_i \cdot \tilde{x}) \right\| < \frac{\varepsilon}{3k}. \tag{65}$$

We then define $n = \max\{j \mid j \in \bigcup_{l=1}^{k} K_l\}$. The complete network combines these approximations through:

$$\|\mathrm{N}_*^\sigma(x) - \mathrm{N}'(x)\| = \left\| \sum_{i=1}^{n} y^{(i)} \sigma(\tilde{R}_i \cdot \tilde{x}) - \sum_{i=1}^{k} (q\sqrt{2} \pm l)_i \sigma(\tilde{w}_i \cdot \tilde{x}) \right\| < \frac{\varepsilon}{3}, \quad \forall x \in \mathcal{K}. \tag{66}$$

**Step (4): Combining Results.** Combining all results, we have:

$$\begin{aligned} \|Y \sigma \left(X_{\mathcal{P}}^\top B^\top C \tilde{x}\right) - f(x)\| &= \|\mathrm{N}_*^\sigma(x) - f(x)\| \\ &< \|\mathrm{N}_*^\sigma(x) - \mathrm{N}'(x)\| + \|\mathrm{N}'(x) - \mathrm{N}^\sigma(x)\| + \|\mathrm{N}^\sigma(x) - f(x)\| \quad (67) \\ &< \varepsilon, \quad \forall x \in \mathcal{K}. \end{aligned}$$

The scalar-output results ($d_y = 1$) extend naturally to vector-valued functions via component-wise approximation. For any continuous $f : \mathbb{R}^{d_x - 1} \to \mathbb{R}^{d_y}$ on a compact domain $\mathcal{K}$, uniform approximation is achieved by independently approximating each coordinate function $f_j$ with scalar networks $\mathrm{N}_{*,j}^\sigma(x)$ satisfying

$$\left\| \mathrm{N}_{*,j}^\sigma(x) - f_j(x) \right\| < \frac{\varepsilon}{\sqrt{d_y}}, \quad \forall x \in \mathcal{K}. \tag{68}$$

The full approximator is then obtained by concatenating the component networks.

$$\mathrm{N}_*^\sigma(x) = \begin{bmatrix} \mathrm{N}_{*,1}^\sigma(x) \\ \vdots \\ \mathrm{N}_{*,d_y}^\sigma(x) \end{bmatrix}, \quad \|\mathrm{N}_*^\sigma(x) - f(x)\| < \varepsilon, \tag{69}$$

$$\mathrm{N}_{*,j}^\sigma(x) = \sum_{i=1}^{n} y_j^{(i)} \sigma(\tilde{R}_i \cdot \tilde{x}), \tag{70}$$

where $y_j^{(i)}$ is the $j$-th row of the $y^{(i)}$. We require that the index sets satisfy $K_i^{(o)} \cap K_j^{(u)} = \emptyset$ for all $o, u, i, j \in \mathbb{N}^+$, where $K_i^{(o)}$ denotes the index set constructed for the $i$-th term approximation in the $o$-th output dimension. Furthermore, each $y^{(j)}$ must have at most one non-zero element across its dimensions. This ensures we achieve uniform approximation by independently handling each output dimension. The proof is complete. $\square$

### E.3 Example of Theorem 7

We present a concrete example with 2D input ($d_x = 2$) and 2D output ($d_y = 2$) to illustrate the universal approximation capability of our architecture. Consider a continuous function $f : [0, 1]^2 \to \mathbb{R}^2$ defined by

$$f(x_1, x_2) = \begin{bmatrix} f_1(x_1, x_2) \\ f_2(x_1, x_2) \end{bmatrix}. \tag{71}$$

Our goal is to construct a module $\mathrm{T}_{*,\mathcal{P}}^\sigma$ such that

$$\left\| \mathrm{T}_{*,\mathcal{P}}^\sigma \left( \begin{bmatrix} x_1 \\ x_2 \\ 1 \end{bmatrix} ; X, Y \right) - f(x_1, x_2) \right\| < \varepsilon. \tag{72}$$

**Step (1): Component-wise Approximation.** For each component $f_i$, there exists a single-hidden-layer neural network $N_i^\sigma(x) = A_i\sigma(W_i x + b_i)$ such that

$$\sup_{x\in[0,1]^2} \|f_i(x) - N_i^\sigma(x)\| < \frac{\varepsilon}{6\sqrt{2}}, \quad i = 1, 2. \tag{73}$$

**Step (2): Rational Perturbation.** We approximate each $N_i^\sigma$ by a rational network $N_i'$:

$$N_1'(x) = (3\sqrt{2} - 2)\sigma(\tilde{w}_1^\top \tilde{x}), \tag{74}$$

$$N_2'(x) = (2\sqrt{2} + 1)\sigma(\tilde{w}_2^\top \tilde{x}), \tag{75}$$

where $\tilde{x} = [x_1 \quad x_2 \quad 1]^\top$, satisfying

$$\sup_{x\in[0,1]^2} \|N_i^\sigma(x) - N_i'(x)\| < \frac{\varepsilon}{6\sqrt{2}}, \quad i = 1, 2. \tag{76}$$

**Step (3): Architecture Realization.** We define a Transformer-like module $N_*^\sigma(x)$ with shared representation:

$$\tilde{R} \approx [\tilde{w}_1 \quad \tilde{w}_1 \quad \tilde{w}_1 \quad \tilde{w}_1 \quad \tilde{w}_1 \quad \tilde{w}_2 \quad \tilde{w}_2 \quad \tilde{w}_2]^\top, \tag{77}$$

$$Y = \begin{bmatrix} \sqrt{2} & \sqrt{2} & \sqrt{2} & -1 & -1 & 0 & 0 & 0 \\ 0 & 0 & 0 & 0 & 0 & \sqrt{2} & \sqrt{2} & 1 \end{bmatrix}, \tag{78}$$

such that

$$N_*^\sigma(x) = \begin{bmatrix} \sum_{i=1}^{8} y_1^{(i)}\sigma(\tilde{R}_i^\top \tilde{x}) \\ \sum_{i=1}^{8} y_2^{(i)}\sigma(\tilde{R}_i^\top \tilde{x}) \end{bmatrix}, \quad \sup_{x\in[0,1]^2} \|N_i'(x) - N_{*,i}^\sigma(x)\| < \frac{\varepsilon}{6\sqrt{2}}. \tag{79}$$

**Step (4): Error Analysis.** The total approximation error satisfies

$$\|f(x) - N_*^\sigma(x)\| \leq \sqrt{\sum_{i=1}^{2} \left(\|f_i - N_i^\sigma\| + \|N_i^\sigma - N_i'\| + \|N_i' - N_{*,i}^\sigma\|\right)^2} \tag{80}$$

$$\leq \sqrt{2 \cdot \left(\frac{\varepsilon}{2\sqrt{2}}\right)^2} = \frac{\varepsilon}{2} < \varepsilon. \tag{81}$$

We argue that in standard ICL, when $y^{(i)} = f(x^{(i)})$ (referred to as meaningfully related) and $\mathcal{V}_y$ satisfies certain conditions, the UAP conclusion still holds. This conclusion relies on the density of $S$, and we provide a concise argument based on Theorem 7.

**Theorem 15.** *Let $\mathcal{T}_{*,\mathcal{P}}^\sigma$ be the class of functions $T_{*,\mathcal{P}}^\sigma$ satisfying Assumption 1, with a non-polynomial, locally bounded, piecewise continuous element-wise activation function $\sigma$, the subscript refers the finite vocabulary $\mathcal{V} = \mathcal{V}_x \times \mathcal{V}_y$, $\mathcal{P} = \mathcal{P}_x \times \mathcal{P}_y$ represents the positional encoding map, and denote a set $S$ as:*

$$S := \mathcal{V}_x + \mathcal{P}_x = \left\{ x_i + \mathcal{P}_x^{(j)} \mid x_i \in \mathcal{V}_x, \ i, j \in \mathbb{N}^+ \right\}. \tag{82}$$

*If $S$ is dense in $\mathbb{R}^{d_x}$, and there exists a subset $Y_0 \subseteq \mathcal{V}_y$ whose (finite) columnwise additive combinations contain the block-diagonal pattern in*

$$\begin{bmatrix} \sqrt{2} & 1 & -1 & 0 & 0 & 0 & 0 & 0 & 0 & \cdots \\ 0 & 0 & 0 & \sqrt{2} & 1 & -1 & 0 & 0 & 0 & \cdots \\ 0 & 0 & 0 & 0 & 0 & 0 & \sqrt{2} & 1 & -1 & \cdots \\ \vdots & \vdots & \vdots & \vdots & \vdots & \vdots & \vdots & \vdots & \vdots & \cdots \end{bmatrix} \tag{83}$$

and $\mathcal{P}_y = 0$, then $\mathcal{T}^\sigma_{*,\mathcal{P}}$ can achieve the UAP. More specifically, for any network $\mathrm{T}^\sigma_{*,\mathcal{P}}$, and for any continuous function $f : \mathbb{R}^{d_x-1} \to \mathbb{R}^{d_y}$ defined on a compact domain $\mathcal{K}$ and any $\varepsilon > 0$, there exist $X \in \mathbb{R}^{d_x \times n}$ and $Y \in \mathbb{R}^{d_y \times n}$ from the vocabulary $\mathcal{V}$, i.e., $x^{(i)} \in \mathcal{V}_x, y^{(i)} \in \mathcal{V}_y$, with some length $n \in \mathbb{N}^+$ such that

$$\left\| \mathrm{T}^\sigma_{*,\mathcal{P}} (\tilde{x}; X, Y) - f(x) \right\| < \varepsilon, \quad \forall x \in \mathcal{K}. \tag{84}$$

We reuse Steps $(1)-(2)$ from the proof of Theorem 7, focusing on understanding the third step. There exists a subset $Y_0 \subseteq \mathcal{V}_y$ whose additive combinations of columns contain the pattern in Eq. (83). The role of this structure is to enable cumulative approximation through additive combinations.

### E.4 Feasibility of UAP under Different Positional Encodings

We also pay attention to more dynamic positional encodings such as RoPE, and are currently exploring appropriate analytical methods for them. Our recent progress on APEs has given us greater confidence in studying RPEs. Our analytical framework mainly relies on achieving density of the set $\sigma(X^\top B^\top C\tilde{x})$, in particular, on the richness of the term $X^\top B^\top C$. (See Lemma 3 and Theorem 7 for supporting arguments).

For RoPE, whose basic formulation is given by equation (16) in [42]

$$q_m^\top k_n = (R^d_{\Theta,m} W_q x_m)^\top (R^d_{\Theta,n} W_k x_n), \tag{85}$$

applying our approach yields

$$\mathrm{T}^\sigma(\tilde{x}, X, Y) = UY\sigma(X^\top B^\top (R^d_{\Theta,1:n})^\top C\tilde{x}). \tag{86}$$

However, since the rotation operation in RoPE acts on distinct two-dimensional subspaces of $d_x$, the induced family $\{B^\top (R^d_{\Theta,j})^\top C\}$ does not generate a dense subset; hence our density-based argument does not directly apply to RoPE. Consequently, our current method cannot be directly applied to prove that RoPE possesses similar approximation properties.

Likewise, other RPEs, such as the one defined in equation (4) of [40],

$$e_{ij} = \frac{x_i W^Q (x_j W^K + a_{ij}^K)^\top}{\sqrt{d_z}}, \tag{87}$$

cannot be analyzed using this approach either. Nevertheless, the encoding formulation in [41],

$$A_{i,j}^{\mathrm{rel}} = \underbrace{E_{x_i}^\top W_q^\top W_{k,E} E_{x_j}}_{(a)} + \underbrace{E_{x_i}^\top W_q^\top W_{k,R} R_{i-j}}_{(b)} + \underbrace{u^\top W_{k,E} E_{x_j}}_{(c)} + \underbrace{v^\top W_{k,R} R_{i-j}}_{(d)}, \tag{88}$$

can be accommodated within our framework and is compatible with the UAP result.

### E.5 Proof of Theorem 8

Before proving Theorem 8, we need to prove the following lemma with the help of the well-known Stone-Weierstrass theorem.

**Lemma 16.** *For any continuous function $f : \mathbb{R}^{d_x} \to \mathbb{R}^{d_y}$ defined on a compact domain $\mathcal{K}$, and for any $\varepsilon > 0$, there exists a network $\mathrm{N}^{\exp}(x) : \mathbb{R}^{d_x} \to \mathbb{R}^{d_y}$ satisfying*

$$\| \mathrm{N}^{\exp}(x) - f(x) \| < \varepsilon, \quad \forall x \in \mathcal{K}, \tag{89}$$

*where $b = 0$ and all row vectors of $W$ are restricted in a neighborhood $B(\omega^*, \delta)$ with any fixed $w^* \in \mathbb{R}^{d_x}$ and radius $\delta > 0$.*

*Proof.* Assume $f(x) = (f_1(x), \cdots, f_{d_x}(x))$. According to Stone-Weierstrass theorem, for any $\varepsilon > 0$, there exist polynomials $P_i(x)$ satisfying

$$\max_{x \in \mathcal{K}} \| P_i(x) - f_i(x) \mathrm{e}^{-w^* \cdot x} \| < \frac{\varepsilon}{2 \max_{x \in \mathcal{K}} \| \mathrm{e}^{w^* \cdot x} \|},$$
$$\Rightarrow \max_{x \in \mathcal{K}} \| P_i(x) \mathrm{e}^{w^* \cdot x} - f_i(x) \| < \frac{\varepsilon}{2}, \quad i = 1, 2, \cdots, d_x. \tag{90}$$

Then we construct a single-layer FNN with exponential activation function to approximate $P_i(x)e^{w^* \cdot x}$. The multiple derivatives of $h(w) := e^{w \cdot x} = \exp(w_1 x_1 + \cdots + w_{d_x} x_{d_x})$ with respect to $w_1, \cdots, w_{d_x}$ are

$$\frac{\partial^{|\alpha|} h}{\partial w^{\alpha}} = \frac{\partial^{|\alpha|} h}{\partial w_1^{\alpha_1} \cdots \partial w_{d_x}^{\alpha_{d_x}}}, \tag{91}$$

where $\alpha \in \mathbb{N}^{d_x}$ represents the index and $|\alpha| := \alpha_1 + \cdots + \alpha_{d_x}$. Actually, the form of multiple derivative $\frac{\partial^{|\alpha|} h}{\partial w^{\alpha}}$ is a polynomial of $|\alpha|$ degree with respect to $x_1, \cdots, x_{d_x}$ times $h(w)$. Hence, each target term $P_i(x)e^{w^* \cdot x}$ can be written as a linear combination of such multiple derivatives of $h(w)$, which allows us to approximate the required partials and thus complete the proof. Moreover, each mixed derivative can be approximated by a finite-difference scheme, which can be implemented using a single hidden layer. □

**Remark 17.** *We give two examples of approximating multiple derivatives of $h(w)$ below.*

$$\begin{aligned} x_1 h(w) &= \left. \frac{\partial h}{\partial w_1} \right|_{w=w^*} \\ &= \frac{h(w^* + \lambda e_1) - h(w^*)}{\lambda} + R_1(\lambda, w^*) \\ &= \lambda^{-1} h(w^* + \lambda e_1) - \lambda^{-1} h(w^*) + R_1(\lambda, w^*), \end{aligned} \tag{92}$$

*and*

$$\begin{aligned} x_1 x_2 h(w) &= \left. \frac{\partial^2 h}{\partial w_1 \partial w_2} \right|_{w=w^*} \\ &= \frac{h(w^* + \lambda(e_1 + e_2)) - h(w^* + \lambda e_1) - h(w^* + \lambda e_2) + h(w^*)}{\lambda} + R_2(\lambda, w^*) \\ &= \lambda^{-1} h((w^* + \lambda(e_1 + e_2)) \cdot x) - \lambda^{-1} h((w^* + \lambda e_1) \cdot x) - \\ &\quad \lambda^{-1} h((w^* + \lambda e_2) \cdot x) + \lambda^{-1} h(w^* \cdot x) + R_2(\lambda, w^*), \end{aligned} \tag{93}$$

*where $e_1 = (1, 0, 0, \cdots, 0)$, $e_2 = (0, 1, 0, \cdots, 0)$ are unit vectors and $R_1(\lambda, w^*)$, $R_2(\lambda, w^*)$ are error terms with respect to $\lambda$ and $w^*$. According to Taylor's theorem, the error terms $R_1(\lambda, w^*) = \lambda \frac{\partial^2 h}{\partial w_1^2} \big|_{w=\xi}$ for some $\xi$ between $w^*$ and $w^* + \lambda e_1$. It is obvious that the partial differential term is uniformly bounded, so the resulting error can be made arbitrarily small by a suitable choice of the parameter $\lambda$. The argument for $R_2(\lambda, W^*)$ is entirely analogous and is therefore omitted; see [61] for further details.*

*Since $\lambda$ is very small and the exponential term $e^{w^* \cdot x}$ only involves the parameters $w^*$, $w^* + e_1$ and $w^* + e_2$, which all lie within a small neighborhood of $w^*$, the desired conclusion can be drawn, and this means we can in fact restrict all row vectors of $W$ to lie within $B(W, \delta)$.*

**Theorem 8** (Formal Version). *Let $\mathcal{T}_{*, \mathcal{P}}^{\sigma}$ be the class of functions $\mathrm{T}_{*, \mathcal{P}}^{\sigma}$ satisfying Assumption 1, with a non-polynomial, locally bounded, piecewise continuous element-wise activation function $\sigma$, the subscript refers to the finite vocabulary $\mathcal{V} = \mathcal{V}_x \times \mathcal{V}_y$, $\mathcal{P} = \mathcal{P}_x \times \mathcal{P}_y$ represents the positional encoding map, and denote a set $S$ as:*

$$S := \mathcal{V}_x + \mathcal{P}_x = \left\{ x_i + \mathcal{P}_x^{(j)} \mid x_i \in \mathcal{V}_x, \ i, j \in \mathbb{N}^+ \right\}. \tag{94}$$

*If the set $S$ is dense in $[-1, 1]^{d_x}$, then $\mathcal{T}_{*, \mathcal{P}}^{\mathrm{ReLU}}$ is capable of achieving the UAP. Additionally, if $S$ is only dense in a neighborhood $B(w^*, \delta)$ of a point $w^* \in \mathbb{R}^{d_x}$ with radius $\delta > 0$, then the class of transformers with exponential activation, i.e., $\mathcal{T}_{*, \mathcal{P}}^{\exp}$, is capable of achieving the UAP.*

*Proof.* For the proof of the ReLU case, we follow the same reasoning as in the previous one, noting that $\mathrm{ReLU}(ax) = a \mathrm{ReLU}(x)$ holds for any positive $a$. In the proof of Theorem 7, we construct a $\mathrm{T}_{*, \mathcal{P}}^{\mathrm{ReLU}}(\tilde{x}; X, Y) \in \mathcal{T}_{*, \mathcal{P}}^{\mathrm{ReLU}}$ to approximate a FNN $A \mathrm{ReLU}(Wx + B)$. Here we can do a similar construction to find another $\tilde{\mathrm{T}}_{*, \mathcal{P}}^{\mathrm{ReLU}}(\tilde{x}; X, Y) \in \mathcal{T}_{*, \mathcal{P}}^{\mathrm{ReLU}}$ to approximate $\lambda A \mathrm{ReLU}\left(\lambda^{-1}(Wx + b)\right)$ as the Step $(2)-(4)$ in Theorem 7, where $\lambda$ is chosen sufficiently large such that the row vectors of $\lambda^{-1} W$ become small enough to ensure that $S = \{x_i + \mathcal{P} \mid x_i \in \mathcal{V}, \ i, j \in \mathbb{N}^+\}$ is dense in $[-1, 1]^{d_x}$ and sufficient for our construction.

For exponential Transformers, by using Lemma 16, we can do the Step $(2)-(4)$ in Theorem 7 again, which is similar to ReLU case. $\qquad\square$

## F  Weakened Assumption and Generalized Conclusions

It is important to note that most of our conclusions remain valid even if Assumption 1 is weakened. Below we outline the reasoning.

In general, we decompose the matrices as follows:

$$Q^\top K = \begin{bmatrix} O_{11} & O_{12} \\ O_{21} & O_{22} \end{bmatrix}, V = \begin{bmatrix} D & E \\ F & U \end{bmatrix}, \tag{95}$$

where $O_{11}, D \in \mathbb{R}^{d_x \times d_x}$, $O_{12}, E \in \mathbb{R}^{d_x \times d_y}$, $O_{21}, F \in \mathbb{R}^{d_y \times d_x}$, and $O_{22}, U \in \mathbb{R}^{d_y \times d_y}$. The attention mechanism can then be computed as:

$$\begin{aligned}
\mathrm{Attn}^\sigma_{Q,K,V}(Z) &= VZM\sigma(Z^\top Q^\top KZ) \\
&= \begin{bmatrix} D & E \\ F & U \end{bmatrix} \begin{bmatrix} X & x \\ Y & 0 \end{bmatrix} \begin{bmatrix} I_n & \\ & 0 \end{bmatrix} \sigma\left( \begin{bmatrix} X^\top & Y^\top \\ x^\top & 0 \end{bmatrix} \begin{bmatrix} O_{11} & O_{12} \\ O_{21} & O_{22} \end{bmatrix} \begin{bmatrix} X & x \\ Y & 0 \end{bmatrix} \right) \\
&= \begin{bmatrix} DX + EY & 0 \\ FX + UY & 0 \end{bmatrix} \sigma\left( \begin{bmatrix} O & (X^\top O_{11} + Y^\top O_{21})x \\ x^\top(O_{11}X + O_{12}Y) & x^\top O_{11}x \end{bmatrix} \right),
\end{aligned} \tag{96}$$

where $O$ represents the matrix $X^\top O_{11}X + X^\top O_{12}Y + Y^\top O_{21}X + Y^\top O_{22}Y$. As a result, we have:

$$\mathrm{T}^\sigma(\tilde{x}; X, Y) = (FX + UY)\sigma\left( (X^\top O_{11} + Y^\top O_{21})\tilde{x} \right), \tag{97}$$

for the case of element-wise activations, and:

$$\mathrm{T}^{\mathrm{softmax}}(\tilde{x}; X, Y) = (FX + UY)\left( \mathrm{softmax}\left( \begin{bmatrix} (X^\top O_{11} + Y^\top O_{21})\tilde{x} \\ \tilde{x}^\top O_{11}\tilde{x} \end{bmatrix} \right) \right)_{1:n}, \tag{98}$$

for the case of softmax activation.

By revisiting the definition of $\mathrm{T}^\sigma$ and $\mathrm{T}^\sigma_*$, and comparing $\mathrm{T}^\sigma$ presented here with those in the preceding section, it is clear that the only distinction lies in the specific matrices involved, and matrix $O_{11}$ and $U$ are non-singular are the only conditions we need. Notably, the proof process for Theorem 6 does not rely on any assumptions, which means the conclusion stated in Section 3 can be further strengthened.

