# OpenReview forum: "Vocabulary In-Context Learning in Transformers: Benefits of Positional Encoding"
_NeurIPS.cc/2025/Conference — NeurIPS 2025 poster_

### Official Review · Reviewer_yZa7 · 2025-06-27

**Clarity:** 3
**Significance:** 2
**Originality:** 3
**Rating:** 4
**Confidence:** 3

**Summary:**

This paper studies the expressive power for the prompts of transformers with absolute positional encodings (PE) when the input prompts are restricted to a vocabulary. Under the assumption on sparseness of $Q,K$ and $V$ matrices of the transformer, the authors start with an equivalence between prompting transformers (without PE) and parametrizing FNNs. Then they show there exists some continuous function $f$ that can not be well approximated by FNNs/transformers if restricting the parameters/prompts to a finite set/vocabulary. But with dense PE, the transformers can break this barrier and approximates any continuous function defined on the compact domain using the prompts.

**Questions:**

Please see weakness.

**Ethical Concerns:**

["NO or VERY MINOR ethics concerns only"]

**Final Justification:**

This paper offers a view that real-valued PEs could add much richer information to inputs from a finite vocabulary, so as to achieve UAP of ICL. Though the main assumption is somewhat idealized (infinite context length), the view itself is interesting to the community. It could encourage future theoretical investigations on learned PEs or structural PEs. Thus I raised the score to 4.

**Limitations:**

yes

**Paper Formatting Concerns:**

No.

**Quality:**

3

**Strengths And Weaknesses:**

Strengths:
1. The writing is easy to follow and most proof sketches are clear.
2. I am not a specialist in approximation theory but I feel it is natural to think about the approximation power under the restriction of vocabulary. Hence the motivation is somewhat clear.

Weakness:
1. One question is on the target $f$. Currently $f$ defined as $\mathcal{K} \to \mathbb{R}^{d_y}.$ As the motivation follows from the fact that all inputs and outputs are from finite vocabularies, I care more about the following target $g: \mathcal{V}_x\to\mathcal{V}_y$ yielded by $f$. (Let's assume $\mathcal{K}=\mathcal{V}_x$ and I'm also happy to assume $\mathcal{V}_x=\mathcal{V}_y=V$). That is, $g(x) = \underset{y\in V}{\arg\max} \ f(x)^\top y$. This is to resemble the scene of greedy decoding. I wonder if UAP still holds for transformers without PE if targets are these functions like $g$.
2. Another question is on the assumption 1. This assumption weakens the expressive power. It would be good if there are some discussions on that.

---

> ### Author Rebuttal · Authors · 2025-07-31
>
> We sincerely appreciate your positive recognition of our work and your thorough discussion of the related issues. After careful consideration of your questions, we would like to offer the following responses:
>
> 1. **[About the target function $f$]** This is a very interesting question, and we are curious about why you considered a function $g(x)$ of this form. After reflecting on this issue, we believe that in most cases, the UAP does not hold in this context. On one hand, the space spanned by all $g(x)$ has a dimension of $|\mathcal{V}_ x| \cdot\dim(\operatorname{span}(\mathcal{V}_ y))$, while on the other hand, the span of ${\sigma(x^{(i)} B^\top C x)}$ has at most dimension $|\mathcal{V}_x|$ (refer to Section 3). It is evident that, in most cases, the space spanned by $g(x)$ has a larger dimension, making it impossible for every $g(x)$ to be contained in the span of ${\sigma(x^{(i)} B^\top C x)}$, and therefore, it is not feasible for all $g(x)$ to be approximated.
>
> 2. **[About the clarification of Assumption 1]** We would like to offer the following explanation for our Assumption 1:
>     - First, the setting where the last row and the last column of the $Q$ and $K$ matrices are all zero is inspired by works [1-3].
>     - Second, our model does not require pretraining, and based on the fact that the probability of a randomly initialized $n$-dimensional matrix being non-singular is $1$, it is acceptable to assume that the matrices $B$, $C$, and $U$ are non-singular.
>     - Furthermore, in Appendix F, we decompose the matrices in the attention mechanism, namely $Q$, $K$, and $V$, and rederive the expression for $\operatorname{Attn}_ {Q,K,V}^\sigma(Z)$. We find that it is sufficient for the submatrix $O_ {11} = B^\top C$ and $U$ to be non-singular. The assumption that $F = 0$ is made for the sake of convenience in proof and computation.
>
> [1] J. Oswald, E. Niklasson, E. Randazzo, J. Sacramento, A. Mordvintsev, A. Zhmoginov, and M. Vladymyrov, “Transformers learn in-context by gradient descent,” in International Conference on Machine Learning, pages 35151–35174. PMLR, 2023.
>
> [2] K. Ahn, X. Cheng, H. Daneshmand, and S. Sra, “Transformers learn to implement preconditioned gradient 397 descent for in-context learning,” in Advances in Neural Information Processing Systems, 2024.
>
> [3] X. Cheng, Y. Chen, and S. Sra, “Transformers implement functional gradient descent to learn non-linear functions in context,” in International Conference on Machine Learning, 2024.

---

> > ### Comment · Reviewer_yZa7 · 2025-08-07
> > **Some follow up**
> >
> > 1. The reason to consider the target $g$ that takes values in the finite vocabulary is that it resembles the real-world scene, where the output of a language model is decoded to be a token ($g$ is defined using greedy decoding with zero temperature). As I understand from your response, PE is still needed to approximate such $g$ in the finite-vocab setting, is that right?
> >
> > 2. I agree with reviewer fhka on the role of PE in this paper. It is true that real-valued PE can provide density (if seq length is long enough) and help with UAP, but I'm not sure if it is relevant. One might argue that a more important aspect of PE is that it adds structural information to the input.

---

> ### Author Response · Authors · 2025-08-09
>
> 1.	We fully understand your motivation for considering such a target function $g$; it indeed aligns well with real-world scenarios. Your interpretation is also correct: to approximate such a function $g$, PEs are indispensable. We have added a corresponding Remark in the revised version of the paper to further elaborate on this point.
>
> 2.	This is indeed an excellent question, and we are grateful for your insightful comments. We want to take this opportunity to offer our understanding and response as follows:
> 	-	First, we would like to emphasize that the conclusion is not detached from practical considerations. In Remark 10, we provide a constructive method for generating dense PEs. This construction ensures the density of the representations $x_ i + \mathcal{P}_ i$ in the space, which theoretically supports the realization of UAP by $\mathcal{T}_{*,\mathcal{P}}^\sigma$. More importantly, this construction guarantees that in practical settings, the desired approximation accuracy can be achieved with only a finite context length.
> 	-	Second, we openly acknowledge that the assumption of “sufficiently long sequences” is somewhat idealized. In practice, models often impose a sequence length limit (e.g., 4096), and the computational complexity of attention is $O(n^2)$. These constraints may limit the attainable approximation accuracy when modeling complex functions.
> 	-	Lastly, we would like to further clarify the structural role of PEs. Due to their injectivity, the same token placed at different positions will yield distinct representations, thereby introducing implicit semantic information about position and order. This structural property is critical for tasks such as language modeling. We have added a discussion on this point in the updated version of the article.
>
> Thank you again for your valuable feedback. We hope our responses address your concerns satisfactorily.

---

### Official Review · Reviewer_ZZKj · 2025-07-01

**Clarity:** 3
**Significance:** 3
**Originality:** 3
**Rating:** 4
**Confidence:** 3

**Summary:**

This paper investigates the role of positional encoding (PE) in the universal approximation property (UAP) of the Transformer architecture. It demonstrates that while a Transformer possesses the UAP under an infinite vocabulary size, there exist functions such as the cosine function that cannot be approximated by any one-layer Transformer without positional encoding. However, once positional encoding is introduced, the one-layer Transformer regains its universal approximation capability.

**Questions:**

Q1: Could you discuss a bit on the relationship with [1], where they also considered the UAP of transformers with PE? Although you mentioned in sec 1.2 that "their implementations allow the internal parameters of the Transformers to vary, which does not fully reflect the characteristics of ICL", I don't fully understand what you mean by "allow the internal parameters of the Transformers to vary".

Q2: In line 118 you mentioned "we do not assume any correspondence between $x^{(i)}$ and $y^{(i)}$, i.e. , $x^{(i)}$ and $y^{(i)}$ are chosen freely", could you explain why this makes sense, since the essence of ICL is for the transformer model to "learn" the relationship of $(x, y)$ pairs from the context information?

Q3: Why is the input to $N^{\sigma}$, namely $x$ in Lemma 3 of dimension $d_{x}-1$ instead of $d_x$?

Q4: Could you explain what is the role of positional encoding $\mathcal{P}_x$ in maintaining the UAP of transformers? I took a brief look at the proof of Theorem 8 and couldn't find whether $\mathcal{P}_x$ is constructed or arbitrarily chosen, so I don't understand how simply adding a $\mathcal{P}_x$ could miraculously recover the UAP for the finite vocabulary setting.

[1] Yun, C., Bhojanapalli, S., Rawat, A. S., Reddi, S. J., & Kumar, S. (2019). Are transformers universal approximators of sequence-to-sequence functions?. arXiv preprint arXiv:1912.10077.

**Ethical Concerns:**

["NO or VERY MINOR ethics concerns only"]

**Final Justification:**

The paper studies an important topic of ICL and makes a first step towards understanding the substantial effect of absolute positional encoding on the UAP of transformers. During the discussion period most of my concerns are addressed, and the authors' argument regarding the relation between $(x_i, y_i)$ seems valid to me.

**Limitations:**

yes

**Quality:**

3

**Strengths And Weaknesses:**

## Strengths
1. The paper studies an important topic of ICL and makes a first step towards understanding the substantial effect of absolute positional encoding on the UAP of transformers.

2. The overall structure of the paper is well-organized and generally easy to follow.

## Weaknesses
1. The paper includes numerous lemmas, propositions, and theorems, some of which appear unnecessary. For instance, Proposition 5 might be omitted from the main text, as its relevance becomes clear only after reading the proof of Lemma 6, which relies on Rolle's Theorem.

2. The paper only studies one-layer transformer and absolute positional encoding, while in practice more dynamic PE such as RoPE are often employed (Although this is indeed a weakness, given the current theoretical literature on APE and ICL of transformers, I understand the difficulty of extension from a theoretical perspective, so I don't expect the authors to fully address the issue).

3. There are some grammar and typo issues, especially in the appendix, e.g.

line 619: "cannot arbitrarily approximated" should be "cannot be arbitrarily approximated"

line 646: "which is contradicts to the Proposition 5" should be "which contradicts Proposition 5"

line 657: "can be view as" should be "can be viewed as"

---

> ### Author Rebuttal · Authors · 2025-07-31
>
> We sincerely appreciate your positive recognition of our work and your thoughtful discussion of the related issues. We have carefully considered your suggestions and would like to offer the following responses:
>
> 1. **[About omitting Proposition 5]** Yes, we agree with your suggestion, and we believe that the revised manuscript has better focus. The detailed proof is still retained in the appendix.
>
> 2. **[About more dynamic positional encodings]** Yes, we are also paying close attention to more dynamic positional encodings such as RoPE and are actively exploring suitable analytical approaches. Our recent progress in studying APEs has given us increased confidence to investigate relative positional encodings as well.
>
>      Our current methodology relies on the density of the matrix $X^\top B^\top C$ in the expression $\sigma(X^\top B^\top C \tilde{x})$ (as shown in the proof of Lemma 3 and Theorem 8 in Appendix B.1). In the case of RoPE, the basic formulation is:
>      $$
>      q_m^\top k_n = (R_{\Theta, m}^d W_q x_m)^\top (R_{\Theta, n}^d W_k x_n) \quad \text{(Eq. 16 in [2])}.
>      $$
>      Applying our method yields:
>      $$
>      \mathrm{T}^{\sigma}(\tilde{x}, X, Y) = UY\sigma(X^\top B^\top (R_{\Theta, m}^{d})^\top C\tilde{x}).
>      $$
>      However, since the RoPE rotation acts independently on different 2D subspaces of the $d_x$-dimensional input, the resulting matrix $B^\top (R_{\Theta, m}^{d})^\top C$ is not dense. Therefore, our current method cannot be directly used to prove that RoPE admits a similar UAP result.
>
>      Similarly, other relative position encodings, such as the one proposed in [3] (Eq. 4):
>      $$
>      e_{ij} = \frac{x_i W^Q (x_j W^K + a_{ij}^K)^\top}{\sqrt{d_z}},
>      $$
>      also cannot be analyzed using our current approach.
>
>      In contrast, the relative positional encoding proposed in [4] is of the following form:
>      $$
>      A_ {i, j}^{\mathrm{rel}} = \underbrace{E_ {x_ i}^\top W_ q^\top W_ {k, E} E_ {x_ j}}_ {(a)} + \underbrace {E_ {x_ i}^\top W_ q^\top W_ {k, R} R_ {i-j}}_ {(b)} + \underbrace{u^\top W_ {k, E} E_ {x_ j}}_ {(c)} + \underbrace {v^\top W_ {k, R} R_ {i-j}} _ {(d)}.
>      $$
>      We find that this encoding can indeed be analyzed using our method and can support the realization of the UAP.
>
> 3. **[About grammar and typo issues]** Thank you for pointing that out. We have made the necessary corrections.
>
> 4. **[About discussion of the relationship with [1]]** We would like to clarify the differences between our work and [1]. We mainly draw from Theorems 2 and 3 of [1] for our understanding:
>
>      - First, the approximation goals differ. [1] focuses on approximating a fixed-length sequence-to-sequence function, $f: \mathbb{R}^{d_x \times n} \to \mathbb{R}^{d_x \times n}$, while our goal is to approximate a general continuous function, $g: \mathcal{K} \to \mathbb{R}^{d_y}$.
>      - Second, for the characterization of approximation, [1] uses the $L^P$ norm, whereas we use the infinity norm. This is a key difference between the two works.
>      - Finally, [1] proves that different Transformers can approximate different target functions, while we show that the same Transformer, with different contextual settings, can approximate different targets. [1] focuses on the approximation ability of the Transformer itself, while our work, grounded in VICL, explores how a fixed-parameter Transformer can approximate various target functions by adjusting its context. Therefore, the sentence in our paper reflects that the Transformer parameters are fixed for different approximation targets, while in [1], the Transformer’s parameters vary for different targets.
>
> 5. **[About $x^{(i)}$ and $y^{(i)}$]** We are excited by your question and have thought carefully about it. We believe that in standard ICL, when $y^{(i)} =f( x^{(i)})$ (which we refer to as meaningfully related) and $\mathcal{V}_ y$ satisfies certain conditions, the conclusion still holds. This relies on the density of the set $S := \mathcal{V}_ x + \mathcal{P}_ x$, and we provide a brief justification based on Theorem 8.
>
>     The main idea in the proof of Theorem 8 is to construct $\mathrm{N}_ * ^\sigma(x)$ to approximate $\mathrm{N}^{\prime}(x)$. Referring to the setup in the sections Implementation Details and Alternative Construction (especially Eq. (82)), we find that the approximation of coefficients such as $(q\sqrt{2}\pm l)_ 1\sigma( \tilde{w}_ 1\tilde{x})$ mainly relies on the sum of the elements in the column vectors of $Y$ corresponding to the approximate location $\tilde{w}_ {1}$. Furthermore, we need some of the $y^{(i)}$ vectors to have column sums that are irrational numbers, positive integers, and negative integers, while the unused positions must have column sums equal to zero.
>
>     In short, if $x^{(i)}$ and $y^{(i)}$ are meaningfully related, and there exist some positions where the column sums of $y^{(i)}$ take on irrational, positive, and negative integer values—while the rest are zero—and $S := \mathcal{V}_ x + \mathcal{P}_ x$ is dense, then the UAP still holds.
>
> 6. **[About dimension $d_x - 1$ in Lemma 3]** The additional dimension is introduced to incorporate the bias term into the model. You can refer to Appendix B, particularly Section B.1, for further details.
>
> 7. **[About the role of positional encoding]** Although the vocabulary is finite, the length of the text is adjustable, and the positional encodings at different positions are dense, which introduces diversity. The realization of the UAP takes advantage of the adjustable length of the text, leveraging the diversity of positional encodings to form more varied combinations for function approximation.
>
>      In Theorem 8, the positional encoding is constructed independently of the target function and the vocabulary. We provide a feasible construction method, with detailed steps available in Appendix E.2 and E.3.
>
> [1] C. Yun, S. Bhojanapalli, A. S. Rawat, S. Reddi, and S. Kumar, “Are transformers universal approximators of sequence-to-sequence functions?” in International Conference on Learning Representations, 2020.
>
> [2] J. Su, M. Ahmed, Y. Lu, S. Pan, W. Bo, and Y. Liu, “Roformer: Enhanced transformer with rotary position embedding,” Neurocomputing, vol. 568, p. 127063, 2024.
>
> [3] P. Shaw, J. Uszkoreit, and A. Vaswani, “Self-attention with relative position representations,” in Annual Meeting of the Association for Computational Linguistics, 2018.
>
> [4] Z. Dai, Z. Yang, Y. Yang, J. Carbonell, Q. Le, and R. Salakhutdinov, “Transformer-xl: Attentive language models beyond a fixed-length context,” in Annual Meeting of the Association for Computational Linguistics, 2019.

---

> > ### Comment · Reviewer_ZZKj · 2025-08-05
> >
> > I thank the authors for the detailed response and most of my questions are solved. It would be beneficial if you could add the discussion on Q2 to the paper or state a new corollary. Overall, I will maintain my score.

---

> ### Author Response · Authors · 2025-08-08
>
> Thank you for your recognition. We have incorporated this part into the latest version of the paper.

---

### Official Review · Reviewer_7CdZ · 2025-07-03

**Clarity:** 3
**Significance:** 3
**Originality:** 3
**Rating:** 4
**Confidence:** 3

**Summary:**

This paper investigated the connection between in-context learning (ICL) in single-layer transformers and universal approximation property (UAP). The main results show that with continuous prompt (e.g., $X$, $Y$), a single-layer attention mechanism can achieve UAP. However, when $X$ and $Y$ are restricted to a discrete set, namely a vocabulary in natural language, the model cannot achieve UAP through ICL alone.  The paper demonstrates that by adding positional encodings, the model can recover the UAP in this discrete setting.

**Questions:**

1. Does Lemma 4 hold for any choice of the number of demonstrations $n$?
2. In Theorem 8, what exactly is the choice of the positional encoding $P$? Is $P$ independent of $(X,Y,x)$ or does it depend on them? Could you clarify it?
  - If $P$ is universal, then $x+P$ lies in a finite set, and by Theorem 7 the model would not achieve UAP.
  - If $P$ depends on the input, this implies that different inputs require different positional encodings, which is not ideal and seems inconsistent with how positional embeddings are used in practice.

3. Have the authors considered the special case where $x^{(i)}\equiv0$ in Theorem 8? Can the model still achieve UAP under this condition? If not what are the roles of $X$ and $P$?

**Ethical Concerns:**

["NO or VERY MINOR ethics concerns only"]

**Limitations:**

yes

**Quality:**

3

**Strengths And Weaknesses:**

Strengths

1. The paper addresses the realistic setting where input prompts come from a discrete vocabulary, reflecting how tokenization works in practice.

2. This work takes attention and ICL as function approximation, establishing an equivalence between FFNs and ICL when treating the input data as function parameters.

Weaknesses

1. In standard ICL, the examples $(x^{(i)},y^{(i)})$ are meaningfully related, and the query feature $x$ relies on the relationship captured by the demonstrations to predict its corresponding $y$. In this work, however, $(x^{(i)},y^{(i)})$ are chosen freely, so the entire matrix $Z$ can be seen as arbitrary function parameters. This breaks the connection to real-world ICL, where the sequential structure and feature-label dependencies are important. The paper would benefit from clarifying this point and discussing how this idealized setting related to the practical ICL scenarios.
2. A key property of ICL is its generalization to new unseen tasks. Discussion on this aspect is missing in the paper.
3. The paper relies on the choice of hidden dimension $k$ and the number of demonstrations $n$ to ensure the results hold, It would strengthen the contribution to include more detailed discussion about how $k$ and $n$ dcale with other parameters, such as dimensions $d_x,d_y$, vocabulary size, function complexity, et al.

---

> ### Author Rebuttal · Authors · 2025-07-31
>
> We sincerely appreciate your positive recognition of our work and your in-depth discussion of the related issues. We have carefully reflected on the shortcomings you pointed out, and would like to offer the following responses:
> 1. **[About $x^{(i)}$ and $y^{(i)}$]** We are excited by your question and have thought carefully about it. We believe that in standard ICL, when $y^{(i)} = f( x^{(i)})$ (which we refer to as meaningfully related) and $\mathcal{V}_ y$ satisfies certain conditions, the conclusion still holds. This relies on the density of the set $S := \mathcal{V}_ x + \mathcal{P}_ x$, and we provide a brief justification based on Theorem 8.
>
>     The main idea in the proof of Theorem 8 is to construct $\mathrm{N}_ * ^\sigma(x)$ to approximate $\mathrm{N}^{\prime}(x)$. Referring to the setup in the sections Implementation Details and Alternative Construction (especially Eq. (82)), we find that the approximation of coefficients such as $(q\sqrt{2}\pm l)_ 1\sigma( \tilde{w}_ 1\tilde{x})$ mainly relies on the sum of the elements in the column vectors of $Y$ corresponding to the approximate location $\tilde{w}_{1}$. Furthermore, we need some of the $y^{(i)}$ vectors to have column sums that are irrational numbers, positive integers, and negative integers, while the unused positions must have column sums equal to zero.
>
>     In short, if $x^{(i)}$ and $y^{(i)}$ are meaningfully related, and there exist some positions where the column sums of $y^{(i)}$ take on irrational, positive, and negative integer values—while the rest are zero—and $S := \mathcal{V}_ x + \mathcal{P}_ x$ is dense, then the UAP still holds.
>
> 2. **[About generalizing to new unseen tasks]** We would like to clarify the intention of our paper. The main conclusion demonstrates that for models satisfying Assumption 1, arbitrary continuous functions over compact domains can be approximated by adapting the context. We do not assume any form of pretraining for the model, which means that every function is essentially unseen to the model.
>
> 3. **[About how $k$ and $n$ scale with other parameters]** We appreciate your suggestion. As you pointed out, the choice of hidden dimension $k$ and the number of demonstrations $n$ indeed plays a crucial role in the validity of our theoretical results. Our current focus is on establishing the feasibility of UAP for a single-layer Transformer under a finite vocabulary and the existence of positional encodings. Therefore, our theoretical framework emphasizes the role and the existence of such positional encodings, rather than providing a full analysis of all parameter interactions.
>     Compared with the feasibility of approximation, analyzing the approximation rate is substantially more difficult. A more precise analysis of approximation rate will be an important direction for future work.
>
> 4. **[About Lemma 4]** We would like to further clarify Lemma 4. This lemma guarantees the existence of $n$, and as discussed earlier, the required number of demonstrations $n$ depends on the target function $f$ and the desired approximation precision $\varepsilon$. Clearly, if $n$ is too small, a good approximation cannot be achieved.
>
> 5. **[About the choice of the positional encoding]** We would like to respond to your question as follows:
>
>     - First, the positional encoding is constructed beforehand. It is universal and independent of any specific context.
>     - Second, Theorem 7 considers whether UAP can be achieved in the absence of positional encoding. Under your proposed setting, if we define $S := \mathcal{V}_ x + \mathcal{P}_ x = \\{x_ i + \mathcal{P}_ x^{(j)} \mid x_ i \in \mathcal{V}_ x,\\ i,\\ j \in \mathbb{N}^+\\}$, then Theorem 7 naturally extends to Theorem 8, and the conclusion that UAP cannot be achieved still holds.
>
>     -  Finally, your second conjecture is actually not valid. In fact, this highlights the value of our basic setting—it is both idealized and aligned with practical modeling scenarios. For more details, please refer to the proof in Appendix E.2 and the illustrative example in Appendix E.3.
>
> 6. **[About the special case in Theorem 8]** We would like to explain Theorem 8 further. The class $\mathrm{T}_ {* ,\mathcal{P}}^\sigma$ consists of functions satisfying Assumption 1, with specific activation $\sigma$ and structural conditions on $\mathcal{V}_ x$, $\mathcal{V}_ y$, and $\mathcal{P}_ x$ (see Theorem 8 for details). For any continuous target function $f$, there exists a specific context and structure such that $\mathrm{T}_ {* ,\mathcal{P}}^\sigma(\tilde{x}; X, Y)$ can approximate $f(x)$. The existence result in the theorem does not exclude the special case where $x^{(i)} \equiv 0$. UAP can still be achieved, but the approximation depends on its position in the context.

---

> > ### Comment · Reviewer_7CdZ · 2025-08-04
> >
> > Thank you for all your responses and explanations. I have decided to maintain my original rating.

---

### Official Review · Reviewer_fhka · 2025-07-14

**Clarity:** 3
**Significance:** 2
**Originality:** 4
**Rating:** 4
**Confidence:** 3

**Summary:**

This theoretical paper investigates the universal approximation property (UAP) of transformers in the specific context of in-context learning (ICL), where the weights of the model are fixed and the input context serves as the parameters. Unlike prior work on the UAP of transformers that assumes continuous, infinite input spaces, this work assumes the input embeddings come from a finite vocabulary of vectors typical in language modeling tasks. By drawing connections to the UAP of single-layer neural networks, the authors show that a single-layer transformer without the input space constraints satisfies UAP, but fails to approximate all functions under the finite-input assumption. However, they prove that adding absolute positional embeddings (PE) restores the UAP, emphasizing the crucial role of PEs in enabling expressivity in this setting.

**Questions:**

In addition to discussing the issues raised above, could you clarify the role of the mask $M$ in Equation (7)? It seems it only zeroes out the last token in the sequence (i.e., the query). What is the motivation for using this particular masking?

**Ethical Concerns:**

["NO or VERY MINOR ethics concerns only"]

**Final Justification:**

The paper presents solid theoretical results. I believe the rebuttal improved it by clarifying the scope and limitations of the findings through additional discussion of the modeling choices and possible extensions of the results. Some idealized assumptions remain, which may limit direct practicality, but with these added discussions, the scope and relevance of the work are much clearer.

**Quality:**

3

**Strengths And Weaknesses:**

The paper offers a fresh perspective on the universal approximation property (UAP) of transformers by focusing on a more realistic in-context learning setup, where inputs are drawn from a finite set of embeddings. This contrasts with previous work that assumes continuous input spaces and helps bridge theoretical insights with practical language modeling scenarios. While I did not follow every detail of the proofs, the sketches provided in the main text and a skim of the appendix suggest that the results are built on neat constructions and connections between neural networks and transformers.

At the same time, some of the assumptions and results appear to be quite strong:

For example, Lemmas 3 and 4 suggest that for any fixed transformer weights, there exists a prompt/context that enables the model to approximate any function, effectively achieving in-context learning even without training. While interesting, this seems far from what is typically observed in practice.

Similarly, the results on the role of positional embeddings (PEs) rely on the assumption of the set of input embeddings + position embeddings forming a dense subset of the space (see Eq. 21). Although Theorem 9 attempts to address this partially, both Theorems 8 and 9 seem to require infinite context lengths to satisfy the density condition, which raises questions about their practical relevance. I believe without this assumption, the UAP won't hold with an argument similar to Theorem 7. From my reading, the role of PE in the proof is essentially to break the finite-input limitation by expanding the support to a dense subset of the space, thereby enabling approximation of any target vector in the space—something not possible with a finite vocabulary alone. This is a clever theoretical construction, but this intuition may not be the reason behind why PEs can help generalization in practical transformer models.

---

> ### Author Rebuttal · Authors · 2025-07-31
>
> We sincerely thank you for your positive recognition of our work and your insightful comments. We have carefully reflected on the limitations you pointed out, and would like to offer the following responses:
>
> 1. **[About Lemmas 3 and 4]** You noted that *"this seems far from what is typically observed in practice."* We agree that most existing theoretical and empirical studies focus on $\mathcal{T}^{\sigma}_{*,\mathcal{P}}$, and we acknowledge that the constructed Transformers in Lemmas 3 and 4 may deviate from practical architectures. However, our intent is to demonstrate that the expressive power of positional encodings enables Transformers to approximate functions with high accuracy. While the construction may be idealized, we believe these lemmas lay a theoretical foundation for understanding how Transformers can achieve high-precision approximation using finite context in practice, and may inspire future analyses of deeper and more practical architectures.
> 2. **[About Positional Encodings]** Your understanding of our density assumption concerning embeddings and positional encodings is entirely correct. Our setting is inspired by the contrast between the finite vocabulary of natural language and the potentially unbounded length of composed sequences, which motivates the need for expressive positional encodings. Although real-world models cannot handle infinitely long inputs, sufficiently long sequences can already allow the model to approximate the desired behavior. While this assumption is admittedly abstract, it serves to highlight the potential role of positional encodings in supporting the generalization capacity of Transformers.
> 3. **[About the mask matrix $M$ in Equation (7)]** The design of the mask matrix $M$ follows previous work [1, 2]. As stated by the authors of [1]: *“Note that the prompt is asymmetric since the label for $x^{(n+1)}$ is excluded from the input. To reflect this asymmetric structure, the mask matrix $M$ is included in the attention.”* Our usage of $M$ follows the same motivation.
>
> [1] K. Ahn, X. Cheng, H. Daneshmand, and S. Sra, “Transformers learn to implement preconditioned gradient descent for in-context learning,” in Advances in Neural Information Processing Systems, 2024.
>
> [2] X. Cheng, Y. Chen, and S. Sra, “Transformers implement functional gradient descent to learn non-linear functions in context,” in International Conference on Machine Learning, 2024.

---

> ### Comment · Reviewer_fhka · 2025-08-05
>
> Thanks for the explanations. I’m still a bit concerned about the relevance of the paper’s take-aways. I understand that theory often needs strong assumptions, yet the insights still have to circle back to the setup we care about.
>
> For lemma 3/4,I’m less worried about the model being close to practice than about what the results seem to say. As I read them, for *any* fixed transformer weights and any target function, you can always find a prompt $(X,Y)$, that makes the transformer approximate that function as closely as you like. That sounds as if an untrained transformer already has in-context learning ability, which feels counterintuitive. The paper should at least discuss this discrepancy and why such a result is relevant, e.g., why pre-training seems necessary for ICL in practice but not in this theory, or what assumptions in the theory might be the reason for this relaxed requirement for ICL.
>
> With the second issue, I feel a similar gap. My main issue is not the assumption of infinite context length; I’m not against an idealized setup if it gives relevant insights. I’m more concerned about whether the conclusion is relevant. The claim is that with real-valued input tokens, we get UAP because we can cover a dense set of functions; with a finite vocabulary, we lose that density. Adding real-valued positional encodings (PE) restores it, but only because the PE vectors are real-valued and, together with an infinite context, give us a dense representation again. First, this density property heavily relies on having an infinite context window, and I don’t see the density argument holding once we limit the context length. Second, it feels odd that the only role of PE here is to inject real-valued vectors to counteract the finite vocabulary (which breaks the density property because of finite-ness), rather than providing something structurally meaningful. I think this also needs a clearer discussion.
>
> I agree with some other modeling issues raised by the reviewers. For example, if the setup does not assume any relationship like $y_i=f(x_i)$ and treats both to be unrestricted parameters, then why model the prompt as a collection of $(x,y)$ pairs at all? Prior ICL works that adopt this modeling use this specific pair-wise relationship for their study.

---

> > ### Author Response · Authors · 2025-08-06
> >
> > Thank you for your response. We would like to provide further clarification.
> >
> > 1. **[About the necessity of pre-training]** We believe that the theoretical conclusions we derive are correct, and we do not consider them to be in conflict with practical observations. Our results merely demonstrate the existence of the context for a fixed  Transformer to approximate a given continuous target function to arbitrary precision, where the required context length depends on the desired accuracy.
> >
> >     We view the role of pre-training as enabling a randomly initialized model to better understand natural language, by guiding the model toward more expressive parameter configurations and potentially reducing the required context length, while still preserving the UAP. While we do not explore the particular architectures or parameter settings induced by pre-training, our existence-based analysis provides a theoretical foundation for extending the study to more general parameter regimes shaped by pre-training.
> >
> >
> > 2. **[About the role of PE]** First, you raised a concern about whether the conclusion is relevant. We would like to emphasize that the conclusion is not detached from practical considerations, as the dense positional encodings required in our analysis can indeed be constructed algorithmically (see Remark 10 in the paper).
> >
> >     Second, your understanding of our argument is highly accurate. We do indeed leverage the density of positional encodings to ensure that $\mathcal{T}_{*,\mathcal{P}}^\sigma$ forms a dense class of functions. Here, we would like to further clarify that, considering the finiteness of the vocabulary in natural language and the potential infiniteness of token sequences, we aim to demonstrate that by introducing appropriate real-valued PEs, high-precision approximation of the target function can be achieved.
> >
> >     Third, your interpretation of the statement “this density property heavily relies on having an infinite context window, and I don’t see the density argument holding once we limit the context length” is entirely valid, and we share the same view.
> >
> >     Lastly, we stress that the role of PE is not merely to inject real-valued vectors to counteract the finite vocabulary. Owing to its injectivity, PE also encodes semantically meaningful information about position and order, which makes its contribution structurally meaningful, rather than merely serving as a mathematical workaround.
> >
> > 3. **[About the relationship between $x_i$ and $y_i$]** We adopt the input-output pair format from ICL tasks when studying the UAP of a fixed Transformer. As shown in the derivation of Eq. (17), the context inputs $X$ and outputs $Y$ play different roles, and UAP can be easily achieved when $X$ and $Y$ are allowed to take values freely in Euclidean space. We then move to a more realistic setting where both $X$ and $Y$ are restricted to a finite vocabulary. In this case, the presence or absence of PE leads to significantly different outcomes. These considerations motivate our modeling of the prompt as a collection of input-output pairs. Furthermore, in standard ICL, when each $y^{(i)} = f(x^{(i)})$ and the output vocabulary $\mathcal{V}_y$ satisfies certain conditions, the conclusion still holds. Please refer to our response to Reviewer 7CdZ, point 1, for further details.

---

> > > ### Comment · Reviewer_fhka · 2025-08-09
> > >
> > > Thanks for the additional explanations.
> > >
> > > I believe all the reviews raised good points about the modeling choices and the relevance of the results and assumptions. I’m not opposed to simplifying assumptions to make the analysis possible, and I think some readers will find the analysis itself interesting. However, the points you addressed and discussed in the rebuttal should be included in the paper to make the scope and limitations of the analysis and results clearer.
> > >
> > > Assuming these additions are included, I’m raising my score to 4.

---

> > > > ### Author Response · Authors · 2025-08-09
> > > >
> > > > We sincerely appreciate your valuable feedback. In the revised version of our paper, we have made the following additions and clarifications:
> > > >
> > > > 1. **[About the mask matrix $M$ in Eq. (7)]** We have added an explanation of the motivation behind adopting this masking design, in order to facilitate readers’ understanding.
> > > > 2. **[About the constructability and role of PE]** In Remark 10, we have emphasized the constructability of positional encodings and further clarified the practical relevance of our conclusion. In Theorem 8, we have added that PEs can partially compensate for the finiteness of the vocabulary, while also carrying semantic information about “position” and “order.”
> > > > 3. **[About standard ICL task]** We have added a corollary stating that in the case of $y^{(i)} = f(x^{(i)})$, if $\mathcal{V}_y$ satisfies certain conditions, the UAP still holds, and we present this result in a formal form.

---

### Comment · Area_Chair_1Gxw · 2025-08-06

Dear Reviewer,

Please review the authors’ rebuttal and finalize your scores. Please write explanations about your updated (or not-updated) scores and submit the Mandatory Acknowledgement. If you have done so, thank you!

Your effort is greatly appreciated for the conference.
Thanks, AC

---

### Note · Authors · 2025-08-13

We sincerely thank all reviewers for their insightful comments and constructive suggestions, which are of great value to improving our work.

In response, we have carefully addressed all concerns raised and made the following revisions in the revised manuscript:

1.	Several reviewers questioned whether our conclusions remain valid under the assumption $y^{(i)} = f(x^{(i)})$ as in standard ICL (7Cdz and ZZkj in the rebuttal stage, fhka in the discussion stage). After careful analysis, we conclude that when $\mathcal{V}_y$ satisfies certain conditions, our results still hold. This has been formally included as a corollary in the revised version.

2.	Many reviewers inquired about the role of PEs (ZZkj in the rebuttal stage, fhka and yZa7 in the discussion stage), specifically whether it is constructed or randomly chosen, and whether it serves purposes beyond ensuring density. We clarify that:
    - The PEs in our setting is relevant and not unique. A constructible example is provided in Remark 10 of the original manuscript.
    - Since different positions have different encodings, the same token in different positions can yield different effects. PEs therefore conveys semantic information about "position" and "order". We have added a dedicated discussion on this point in the revised Discussion section.

3.	Reviewer fhka asked for the motivation behind the design of the mask matrix $M$. This has now been explicitly stated in the revised manuscript.

4.	Reviewer yZa7 noted that Assumption 1 might weaken our conclusions and suggested further discussion. We have added a remark explaining its necessity and reasonableness.

5.	Reviewer ZZkj suggested omitting Proposition 5 from the main text. We agree with this suggestion; the detailed proof is retained in the appendix.

6.	Reviewer ZZkj suggested discussing more dynamic PEs, such as RoPE. we have included a discussion in the Discussion section based on the proposed method in the revised manuscript.

7.	Reviewer ZZKi pointed out some grammar and typo errors, which have been corrected in the revised manuscript.

---

### Decision · Program_Chairs · 2025-09-17

**Decision:**

Accept (poster)

**Comment:**

This work considers the representation capacity of transformers in in-context learning, ie, fix transformer weights while varying the context to represent functions. It considers the setting where the context is represented by tokens from a finite vocabulary. The main result is that single-layer transformers without positional encoding do not have the universal approximation property, while it is possible with positional encoding.

The work provides an interesting perspective on the capacity of transformers in in-context learning. The analysis and results are solid. In the discussions, the authors have clarified the scope and meaning of the results and some technical details, after which all reviewers leaned towards acceptance. The work is recommended for acceptance, and the authors are suggested to include the revisions in the final version if accepted.